# Cyclophilin A enables specific HIV-1 Tat palmitoylation and accumulation in uninfected cells

Christophe Chopard[1], Phuoc Bao Viet Tong[1], Petra Tóth[2], Malvina Schatz[1], Hocine Yezid[1], Solène Debaisieux[1], Clément Mettling[3], Antoine Gross[1], Martine Pugnière[4], Annie Tu [2], Jean-Marc Strub[5], Jean-Michel Mesnard[1], Nicolas Vitale[2,6] & Bruno Beaumelle[1]

Most HIV-1 Tat is unconventionally secreted by infected cells following Tat interaction with phosphatidylinositol (4,5) bisphosphate ($PI(4,5)P_2$) at the plasma membrane. Extracellular Tat is endocytosed by uninfected cells before escaping from endosomes to reach the cytosol and bind $PI(4,5)P_2$. It is not clear whether and how incoming Tat concentrates in uninfected cells. Here we show that, in uninfected cells, the S-acyl transferase DHHC-20 together with the prolylisomerases cyclophilin A (CypA) and FKBP12 palmitoylate Tat on Cys31 thereby increasing Tat affinity for $PI(4,5)P_2$. In infected cells, CypA is bound by HIV-1 Gag, resulting in its encapsidation and CypA depletion from cells. Because of the lack of this essential cofactor, Tat is not palmitoylated in infected cells but strongly secreted. Hence, Tat palmitoylation specifically takes place in uninfected cells. Moreover, palmitoylation is required for Tat to accumulate at the plasma membrane and affect $PI(4,5)P_2$-dependent membrane traffic such as phagocytosis and neurosecretion.

[1] IRIM, UMR 9004, Université de Montpellier-CNRS, 1919 Route de Mende, 34293 Montpellier, France. [2] INCI, UPR 3212 CNRS, 5 rue Blaise Pascal, 67084 Strasbourg, France. [3] IGH, UPR 1142 CNRS, 141 Rue de la Cardonille, 34396 Montpellier, France. [4] IRCM, INSERM U 1194, 208 Rue des Apothicaires, 34298 Montpellier, France. [5] CNRS, IPHC UMR 7178, Université de Strasbourg, 67000 Strasbourg, France. [6] INSERM, 75654 Paris Cedex 13, France. Correspondence and requests for materials should be addressed to P.Tót. (email: petra.toth@inci-cnrs.unistra.fr) or to B.B. (email: bruno.beaumelle@irim.cnrs.fr)

HIV-1 Tat enables robust transcription from HIV-1 LTR. This small basic protein is strictly required for viral gene expression and HIV-1 virion production[1]. But Tat can also be secreted by infected cells using an unconventional pathway[2]. This secretion is based on the strong and specific interaction of Tat with phosphatidylinositol (4,5) bisphosphate (PI(4,5)P$_2$), a phosphoinositide that is specifically concentrated on the inner leaflet of the plasma membrane[3] and enables Tat recruitment at this level. Tat export is very active since ~2/3 of Tat are secreted by infected T-cells[4]. Consistently, a Tat concentration in the nanomolar range has been detected in the sera of HIV-1 infected patients[5–7]. Circulating Tat acts as a viral toxin. Tat is endocytosed by most cell types[8] and, once in the endosome, low pH triggers unmasking of Trp11, enabling membrane insertion that culminates with Hsp90-assisted Tat translocation to the cytosol[9,10]. Incoming Tat induces a variety of cell responses[11]. Indeed, Tat is able to modify the expression of cellular genes[12], some of them being involved in cell transformation and leading to the development of HIV-1 associated cancers[13]. Tat is also a key regulator of HIV-1 latency[14].

Palmitoylation (or S-acylation) is the thioester linkage of a palmitate (the most abundant fatty acid) to a cysteine, resulting in membrane tethering. In mammals, a family of 23 protein acyl transferases that share a conserved DHHC sequence in their active site has been identified[15].

HIV-1 infected patients suffer from defects in phagocytosis[16] and cardiac repolarization[17]. They also present various neuro-cognitive disorders[18]. We accordingly showed that, in target cells such as macrophages, neurons and myocytes, incoming Tat binds to PI(4,5)P$_2$ and severely inhibits cell machineries that rely on protein recruitment by this phosphoinositide, i.e., phagocytosis, neurosecretion and key cardiac potassium channels[19]. To this end, Tat prevents cdc42 recruitment at the phagocytic cup in macrophages thereby inhibiting phagocytosis[20]. In neuroendocrine cells, Tat impairs the recruitment of annexin-2 to the exocytic sites, resulting in neurosecretion inhibition[21]. In myocytes, Tat accelerates hERG and KCNE1/KCNQ1 deactivation, thereby increasing action potential duration[22].

Intriguingly, especially in the phagocytosis case, minute doses of Tat (~0.2 nM) only were necessary to be effective. This observation raises two questions. How can such small doses of Tat be inhibitory while plenty of PI(4,5)P$_2$ (~ 10 μM[23]) is present within cells? And how is it possible for Tat to perturb PI(4,5)P$_2$ mediated protein recruitment while it should quickly abandon PI(4,5)P$_2$ to cross the plasma membrane for secretion?

We here propose a response to both issues: Tat is palmitoylated in target cells, such as T-cells, macrophages and neurons. We found that Tat is specifically palmitoylated on Cys31 by the S-acyl transferase DHHC-20. Tat palmitoylation prevents Tat secretion and enables Tat accumulation on PI(4,5)P$_2$ at the plasma membrane thereby allowing this viral toxin to severely interfere with PI(4,5)P$_2$-dependent membrane traffic. This result in turn raises the question of how can infected T-cells secrete Tat so actively. Indeed, it is difficult to reconcile the efficiency of this export with Tat palmitoylation that should prevent it. In fact, the viral Gag protein interacts with cyclophilin A (CypA), resulting in its encapsidation[24]. We found that HIV-1 budding essentially depletes cells in CypA and, because CypA is required for Tat palmitoylation, this process is thereby inhibited in infected cells. HIV-1 thus uses an elaborate mechanism to efficiently ensure both Tat secretion by infected T-cells and Tat retention on PI(4,5)P$_2$ in uninfected cells.

## Results

**Incoming HIV-1 Tat is palmitoylated in various cell types.** We used His$_6$-tagged Tat and the click chemistry technique[25] to examine whether exogenous Tat can be palmitoylated in various cell lines, i.e., human T-cells (Jurkat), macrophages (RAW 264.7) and neurosecretory cells (PC12 cells). To this end, cells were incubated with Tat-His$_6$ and 17-octadecanoic acid (17-ODYA), a palmitate analog with a terminal alkyne group. Tat became labeled with 17-ODYA in all these cell lines (Fig. 1a), indicating that most cell types are able to palmitoylate incoming Tat.

Neurons are an important target for HIV-1 Tat[21] and to study in detail Tat palmitoylation, we transiently transfected Tat-FLAG in the neurosecretory cell line PC12 and detected palmitoylation using both the click-chemistry approach[25] and the acyl-biotin exchange technique (ABE) that is based on the replacement of the protein-acyl chain by a biotin. This exchange relies on the use of hydroxylamine that selectively cleaves thioester bonds[26]. Both the click chemistry (Fig. 1b) and the ABE (Fig. 1c) techniques showed that transfected Tat is palmitoylated. Addition of 2-bromopalmitate (2-BP), a well-established palmitoylation inhibitor[27] prevented Tat palmitoylation (Fig. 1b). The sensitivity of Tat-acyl chain bond to hydroxylamine (Fig. 1c) showed that this bond is a thioester bond[26] and thus that the acyl chain is attached to at least one of the seven Cys of Tat.

**Tat is specifically palmitoylated on Cys31.** Tat Cys are located in position 22, 25, 27, 30, 31, 34, and 37 and, to identify the palmitoylated Cys, we individually mutated each of them in Ser. A double mutant Tat-C30S-C31S was also generated because, in the case of Cys doublets, the mutation of one of the Cys can sometimes aberrantly lead to palmitoylation of the remaining Cys[28]. We then examined the capacity of these mutants to be palmitoylated in PC12 cells. All mutants were expressed to similar levels, but those devoid of Cys31 only failed to be palmitoylated (Fig. 2a), indicating that Tat is palmitoylated on Cys31. Mass spectrometric analysis of immunoprecipitated Tat-FLAG using nano LC-MS/MS confirmed that Tat is palmitoylated on Cys31 (Supplementary Table 1).

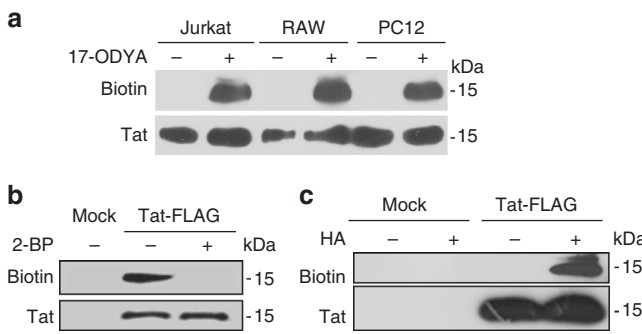

**Fig. 1** Tat is palmitoylated in T-cells, macrophages and neuron precursors. **a** T-cells (Jurkat), macrophages (RAW 264.7) or neuron precursors (PC12 cells) were labeled overnight with 17-ODYA and Tat-His$_6$ in lipid-free medium before cell lysis, Tat-His$_6$ purification, biotin labeling of 17-ODYA using click chemistry and SDS-PAGE. The blot was first incubated with avidin-peroxidase to detect biotin, then with anti-Tat antibodies. **b** PC12 cells were transfected with Tat-FLAG before overnight labeling with 17-ODYA, anti-FLAG immunoprecipitation, and click chemistry. When indicated 100 μM 2-bromopalmitate (2-BP) was present during labeling with 17-ODYA. **c** PC12 cells were transfected with Tat-Flag before anti-Flag immunoprecipitation and acyl-biotin exchange. Hydroxylamine (HA) treatment is used to remove the acyl chain and enables to replace it with biotin for palmitoylation detection. Representative data from $n = 3$ experiments

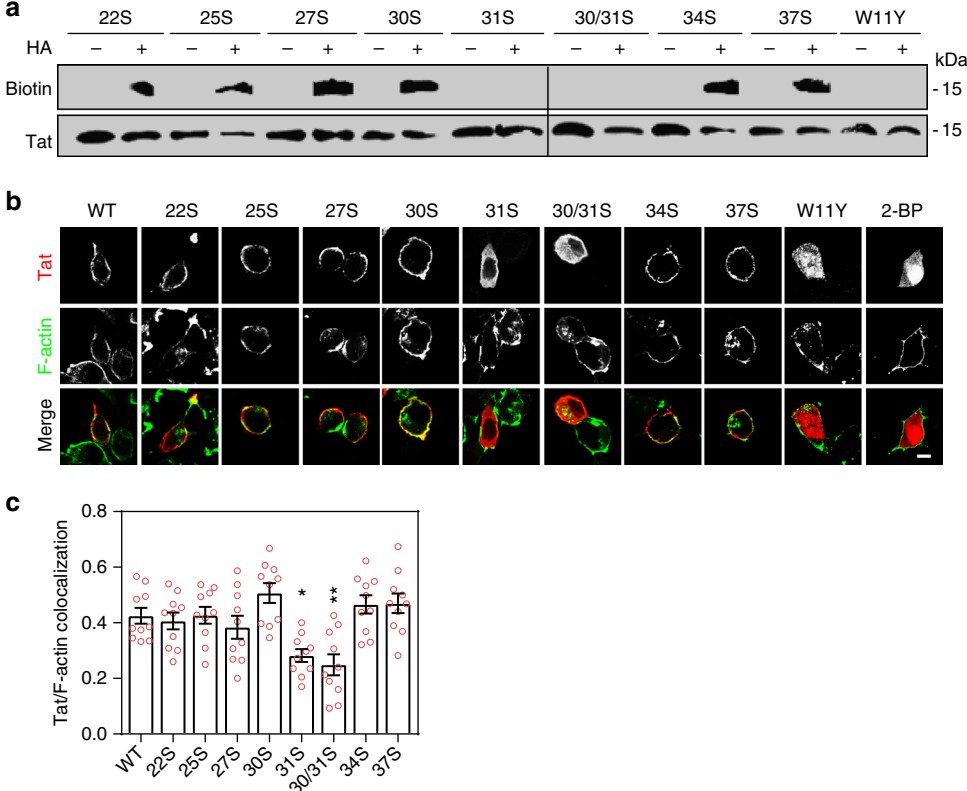

**Fig. 2** Tat palmitoylation takes place on Cys31 and requires PI(4,5)P$_2$ binding. **a** PC12 cells were transfected with the indicated Tat-FLAG-CXXS or Tat-W11Y mutant before anti-FLAG immunoprecipitation, acyl-biotin exchange (HA, hydroxylamine), western blot and biotin then Tat detection. Images from two blots were assembled. Films with the same exposure time were used and the dashed line marks the separation between them. **b** Tat-transfected PC12 cells were processed for detection of F-actin using fluorescent phalloidin and Tat using immunofluorescence. When indicated, cells transfected with WT Tat were treated with 100 μM 2-BP for 5 h before fixation. Representative median confocal sections are shown. Bar, 5 μm. **c** plasma membrane localization of Tat-mutants was evaluated by quantifying Tat/F-actin colocalization using confocal images from $n = 10$ cells and Mander's coefficient calculation (mean ± SEM). The significance of differences with WT Tat was assessed using one-way ANOVA (*$p < 0.05$; **$p < 0.01$). Similar microscopy data were obtained using primary neurons (Supplementary Fig. 3)

To examine whether the C31S mutation affected Tat structure we first assessed the capacity of Tat-C31S to transactivate a co-transfected LTR-driven luciferase. The results showed that transfected Tat-C31S displayed native Tat transactivation capacity (Supplementary Fig. 1), indicating that the C31S mutation does not affect Tat structure and reactivity. When the same experiment was performed using extracellular recombinant Tat-C31S, no difference was observed either. Hence, the C31S mutation did not impair the capacity of Tat to enter cells. We also examined, using liposomes and isothermal calorimetry (ITC)[4], whether this mutation affects Tat affinity for PI(4,5)P$_2$ and found that Tat-C31S, with a Kd of $8 \pm 5$ nM (mean ± SEM, $n = 3$) has essentially the same affinity for PI(4,5)P$_2$ as WT Tat (Kd = 55 ± 20 nM; Supplementary Fig. 2). Collectively, these results show that the C31S mutation prevents palmitoylation but does not significantly affect Tat folding or transactivation activity. They are consistent with the observation that HIV-1 clade C present in India and that has a Tat-31S is essentially as virulent as other clades regarding its capacity to multiply in peripheral blood mononuclear cells (PBMCs)[29].

We then examined the subcellular localization of Tat Cys mutants in PC12 cells. All mutants except Tat-C31S and Tat-C30S-C31S localized to the plasma membrane, while a large fraction (~60%) of Tat-C31S and Tat-C30S-C31S accumulated in the cytosol. Similar data were obtained when palmitoylation was prevented using 2-BP (Fig. 2b). This observation not only indicated that Tat palmitoylation is required for its stable

association with the plasma membrane, but also that a large fraction of Tat is palmitoylated in PC12 cells. Similar results were obtained using primary hippocampal neurons (Supplementary Fig. 3). Altogether, this first set of data indicates that, in neuroendocrine cells as well as primary neurons, a significant pool of Tat is palmitoylated on Cys31 and that this modification is important for plasma membrane localization. We examined next whether incoming Tat-C31S became palmitoylated in Jurkat cells. This was not the case (Supplementary Fig. 4) and, since Tat-C31S is not affected in its internalization pathway (Supplementary Fig. 1) this result confirms that Cys31 is the only palmitoylated residue in Tat, whether Tat-producer (T-cells) or Tat-target cells (neuroendocrine cells) are used.

**Tat palmitoylation requires PI(4,5)P$_2$ binding**. Tat-W11Y poorly binds PI(4,5)P$_2$[4], while showing native transactivation activity, indicating that this mutation does not affect Tat conformation[9]. This mutant, as observed before in T-cells[4], is cytosolic and/or nuclear in PC12 cells (Fig. 2b) and Tat-W11Y failed to be palmitoylated (Fig. 2a). Hence, PI(4,5)P$_2$ binding appears as a prerequisite for Tat palmitoylation, indicating that this modification takes place at the plasma membrane.

**Tat palmitoylation is performed by DHHC-20**. DHHC enzymes are membrane proteins that can acylate proteins once the substrate has reached the appropriate membrane[30]. In agreement

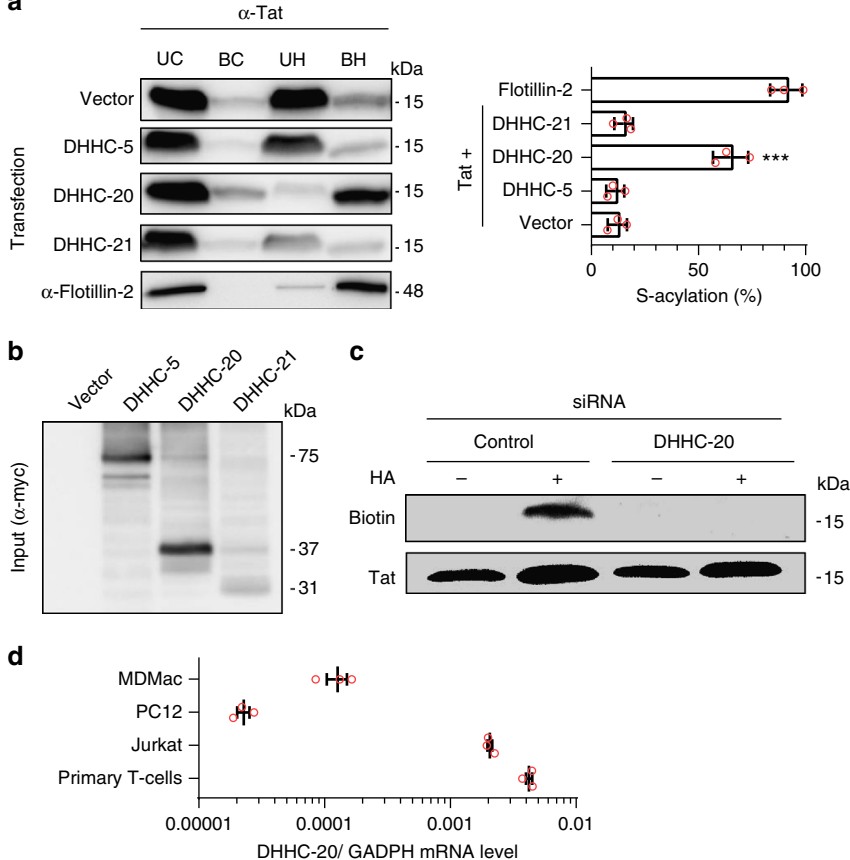

**Fig. 3** Tat palmitoylation is performed by DHHC-20. **a** HEK 293 T cells were cotransfected (1/5) with the indicated myc-tagged human DHHC and Tat. Tat palmitoylation was then assessed using the acyl-RAC technique, UC unbound control, BC bound control, UH unbound hydroxylamine, BH bound hydroxylamine. Palmitoylated Tat is present in the BH fraction. Palmitoylation was calculated as BH/(BH + UH)-BC/(BC + UC), and flotillin-2 was used as a positive control. The graph shows mean ± SEM of $n = 3$ independent experiments, ***$p < 0.001$ (one-way ANOVA). **b** DHHC-myc overexpression level was assessed using anti-myc western blot. DHHC-5, -20, and -21 migrated at 75, 37 and 31 kDa, respectively. Transfection efficiency was 60–70%. **c** PC12 Cells were cotransfected with Tat-FLAG and the indicated siRNA before detecting Tat palmitoylation using acyl-biotin exchange. The efficiency of siRNA-mediated silencing is shown in Supplementary Fig. 6. **d** The RNA from monocyte-derived macrophages (MDMac), PC12, Jurkat or human primary CD4+ T-cells was extracted before quantification of DHHC-20 and GAPDH mRNAs using qRT-PCR. Results are expressed as DHHC-20/GAPDH ratio. Results for all DHHCs are shown in Supplementary Fig. 7. Mean ± SEM, $n = 3$ independent experiments

with previous observations[31] we found that, among the 23 DHHC proteins, DHHC-5 and DHHC-20 only localized to the plasma membrane in PC12 cells (Supplementary Fig. 5). Since Tat is palmitoylated at this level, we first examined using an over-expression approach which of DHHC-5 or DHHC-20 favors Tat palmitoylation, using DHHC-21 as control. We did not use PC12 cells for this experiment because Tat palmitoylation is already very efficient in this cell type (Fig. 2b). We used HEK 293 T cells that were cotransfected with Tat and myc-tagged human DHHC-5, -20 or -21. Exogenous DHHC-5 and DHHC-20 were expressed to the same level while DHHC-21 expression was ~3-fold lower (Fig. 3b). Tat palmitoylation was assessed using the acyl-RAC technique, a variation of the ABE technique that enables easier quantification. Flotillin-2 which is known to be stably palmitoy-lated was used as positive control[32]. The efficiency of overexpressed-Tat palmitoylation in HEK cells (~15%) was not affected by cotransfection with DHHC-5 or DHHC-21, while it reached 65% upon DHHC-20 coexpression (Fig. 3a). This result indicated that Tat is palmitoylated by DHHC-20. To confirm this finding, we used siRNAs against rat DHHC-20 that silenced its expression by ~80% in PC12 cells according to qRT-PCR quan-tification (Supplementary Fig. 6). Silencing DHHC-20 led to a complete inhibition of Tat palmitoylation, while the control siRNA had no effect (Fig. 3c). Hence, Tat palmitoylation is

performed by DHHC-20, an enzyme that is expressed in T-cells, macrophages and PC12 cells (Fig. 3d for DHHC-20 and Sup-plementary Fig. 7 for all DHHCs). In agreement with previous observations, DHHC localization can be cell type-dependent[15], and in fact DHHC-20 seems to be the only DHHC that localizes to the plasma membrane in primary T-cells, while DHHC-5-EGFP was observed at the Golgi apparatus in these cells (Sup-plementary Fig. 8).

**Tat palmitoylation increases its affinity for PI(4,5)P$_2$.** We then assessed whether palmitoylation affects Tat affinity for PI(4,5)P$_2$. To this end, we obtained a Tat devoid of Cys except Cys31 (Tat-C31 only termed Tat-C31O) that was labeled with palmitate. We had to use this strategy because it was not possible to specifically label Cys31 with palmitate when the other Cys are present. MALDI-TOF/TOF analysis indicated that Tat-C31O was acylated by a single palmitate with an efficiency of 47 ± 8% (Supple-mentary Fig. 9). Using PI(4,5)P$_2$-containing liposomes and Sur-face Plasmon Resonance we observed that the affinity for PI(4,5)P$_2$ of Tat-C31O (Kd = 0.31 ± 0.02 nM) was the same as that of native Tat[4], while Tat-C31O-Palm showed a Kd of 0.18 ± 0.02 nM, suggesting that palmitoylation increases Tat affinity for PI(4,5)P$_2$. Since only half of Tat-C31O was palmitoylated these

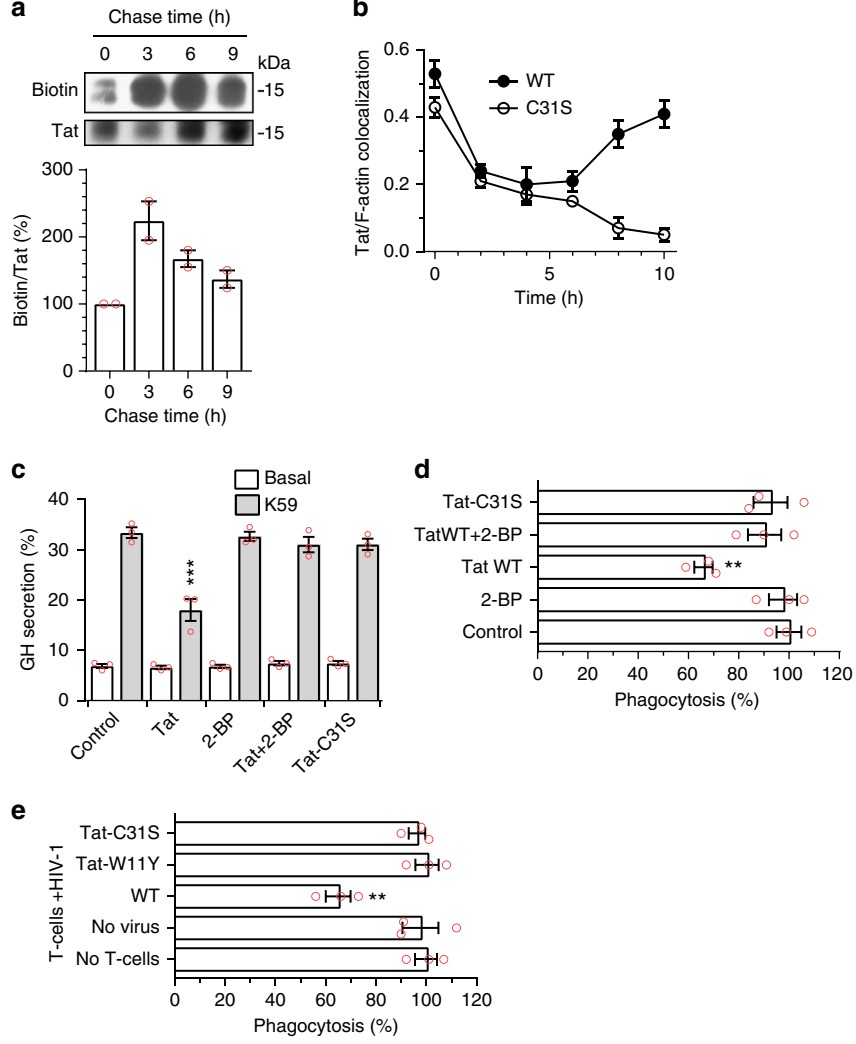

**Fig. 4** Palmitoylation is a stable modification that enables Tat accumulation at the plasma membrane and Tat-mediated inhibition of $PI(4,5)P_2$-dependent membrane traffic. **a** PC12 cells were labeled overnight with Tat-His$_6$ and 17-ODYA before chasing for the indicated times, cell lysis, Tat-His$_6$ purification and click chemistry. Results from a representative experiment are shown. The graph (mean ± SEM of 2 independent experiments) shows Tat palmitoylation efficiency as the biotin/Tat ratio (% of the t0 value) as a function of time. The increase of labeling during the first 3 h of chase is probably due to residual intracellular 17-ODYA. **b** PC12 cells were labeled with Tat (WT or C31S) at 4 °C before washing, chasing for the indicated times and Tat staining by immunofluorescence with F-actin labeling using fluorescent phalloidin. Tat membrane localization was evaluated by quantifying Tat/F-actin colocalization using confocal images from 50 < n < 100 cells and Mander's coefficient calculation. Mean ± SEM. Representative images are in Supplementary Fig. 10. **c** PC12 cells were transfected with human growth hormone (GH) then treated for 5 h with 100 µM 2-BP, 20 nM Tat WT or Tat-C31S as indicated. GH secretion was then triggered using 59 mM K$^+$ (K59) and quantified by ELISA. Mean ± SEM (n = 3). **d** Macrophages (MDMs) were treated for 3 h with 100 µM 2-BP, 5 nM Tat or Tat-C31S as indicated before assaying phagocytosis of IgG-coated 3 µm latex beads. Extracellular beads were stained before cell fixation and examination using a fluorescence microscope. **e** CD4 + T cells were purified, stimulated then infected with a T-tropic (NL4.3) virus, bearing Tat-WT, -W11Y or -C31S as indicated, then added to Transwells into wells containing autologous MDMs. FcγR-mediated phagocytosis by MDMs was assayed after 8 days of co-culture. At this time 20–35% of T cells and 0% of macrophages were infected. Data in panels **d** and **e** are mean ± SEM of three independent experiments (counting n > 100 cells for each). The significance of differences with controls was assessed using ANOVA, one-way (**d**, **e**) or two-way (**c**) (***$p < 0.001$; **$p < 0.01$)

results indicate that palmitoylation increases Tat affinity for PI $(4,5)P_2$ by ~5-fold.

**Tat palmitoylation is a stable modification**. To examine the stability of Tat palmitoylation, PC12 cells were labeled overnight with Tat-His$_6$ and 17-ODYA before a chase. Tat palmitoylation showed a half-life of ~7.3 h ($R^2 = 0.985$; Fig. 4a), indicating that this is a stable modification.

**Palmitoylation enables Tat accumulation at the plasma membrane**. We then assessed the effect of Tat palmitoylation on the

plasma membrane recruitment of incoming Tat. To this end cells were labeled with recombinant Tat at 4 °C before washing and chasing at 37 °C. Under these conditions, Tat is first observed at the plasma membrane, then in endocytic vesicles from which it escapes to reach the cytosol, and it is finally recruited at the plasma membrane by $PI(4,5)P_2$[21]. This relocalization to the plasma membrane can be followed using Tat/F-actin colocalization, and is clearly observed for WT Tat at late chase time points (8–10 h; Fig. 4b; representative images are in Supplementary Fig. 10). In sharp contrast, Tat-C31S is not recruited to the plasma membrane after 8–10 h of chase. In fact, Tat-C31S disappears from the cell at late time points. These results obtained

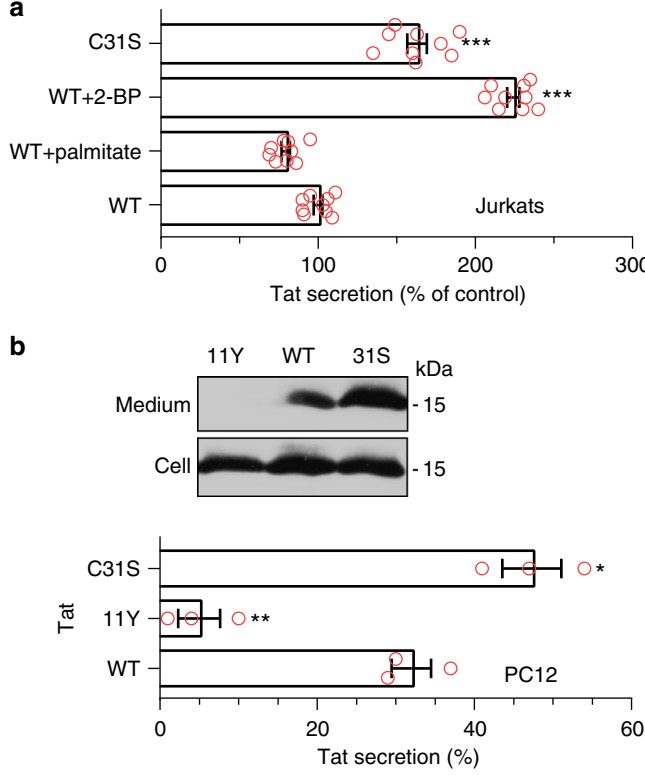

**Fig. 5** Tat palmitoylation inhibits its secretion. **a** Jurkat cells were transfected with Tat or Tat-C31S before treatment for 3 h with 50 μM 2-BP or palmitate as indicated and assaying Tat secretion by ELISA (Mean ± SEM, $n = 3$ independent experiments performed in triplicate). **b** PC12 cells were treated similarly before assaying Tat secretion using immunoprecipitation and Western blotting. Results from a representative experiment are shown together with the quantification (Mean ± SEM, $n = 3$). The significance of differences with controls was assessed using one-way ANOVA (***$p < 0.001$; **$p < 0.01$; *$p < 0.05$)

with incoming Tat are consistent with data from Tat-transfected cells (Fig. 2b) and indicate that Tat palmitoylation enables stable Tat association with PI(4,5)P$_2$.

**Palmitoylated Tat inhibits PI(4,5)P$_2$-dependent traffic.** If Tat palmitoylation is required for stable Tat association with PI(4,5)P$_2$ it would be needed for Tat to affect PI(4,5)P$_2$-dependent membrane traffic. We examined this issue using both 2-BP treatment and Tat-C31S that cannot be palmitoylated (Fig. 2a and Supplementary Fig. 4). We first assessed whether palmitoylation is involved in Tat capacity to inhibit neurosecretion. To this end, PC12 cells were transfected with human growth hormone (GH) before treatment with extracellular Tat. In this assay, GH is incorporated in secretion granules that, upon stimulation, fuse with the plasma membrane in a process that recapitulates neuromediator secretion, a PI(4,5)P$_2$-dependent process[21]. Tat strongly inhibited GH release (−40%), as observed before[21], while 2-BP had no effect. Both treatment with 2-BP and the C31S mutation prevented Tat inhibitory effect on neurosecretion (Fig. 4c). Identical data were obtained using primary chromaffin neurorendocrine cells (Supplementary Fig.11). These data showed that palmitoylation is required for Tat to inhibit neurosecretion.

We also examined whether Tat palmitoylation is needed for Tat to inhibit FcγR-mediated phagocytosis. Tat inhibits this key phagocytic process by preventing the PI(4,5)P$_2$-mediated recruitment of cdc42 to the phagocytic cup[20]. Monocyte-derived human primary macrophages (MDMs) were treated with Tat, Tat-C31S

or 2-BP before assaying the phagocytosis of IgG-coated latex beads. Tat inhibited phagocytosis by ~ 35% in this assay whereas 2-BP had no significant effect. Either 2-BP or the C31S mutation blocked Tat inhibitory effect on phagocytosis (Fig. 4d). It was important to validate these findings in the context of infection. To this end, primary CD4$^+$ T-cells were infected with T-tropic viruses bearing either Tat WT, Tat-W11Y or Tat-C31S that were then added on autologous MDMs using a Transwell configuration. After 8 days, macrophage FcγR-mediated phagocytic activity was assayed. At this time, T-cells are efficiently infected, while macrophages remain uninfected. Tat secretion by T-cells into the medium is responsible for phagocytosis inhibition[20] and the W11Y mutation that impairs both Tat secretion[4] and entry into cells[9] prevents Tat from affecting phagocytosis[20]. The C31S mutation also prevented Tat effect on phagocytosis in this coculture assay (Fig. 4e). Since Tat-C31S is actively secreted (Fig. 5a) and efficiently enters cells (Supplementary Fig.1) these results confirmed, in primary cells, that Tat palmitoylation enables it to perturb phagocytosis.

Altogether, these data showed that Tat palmitoylation is needed for Tat to inhibit PI(4,5)P$_2$-dependent membrane traffic processes such as neurosecretion and phagocytosis.

**Tat palmitoylation inhibits its secretion.** Since palmitoylation increases Tat affinity for PI(4,5)P$_2$ and stabilizes Tat association with the plasma membrane (Fig. 4b), it should affect Tat secretion. We examined Tat secretion in two cell types, T-cells (Jurkat, producer cells) and PC12 (target cells), using different assays[33]. Tat secretion by Jurkat cells (~15% in 6 h[4]) was weakly inhibited in the presence of palmitate (−20%), and strongly enhanced by 2-BP (+120%) or by introduction of the C31S mutation (+60%) (Fig. 5a). Similar data were obtained using PC12 cells (Fig. 5b). Hence, although Tat-C31S is more cytosolic than WT Tat (Fig. 2b; Supplementary Fig. 3), it is more efficiently secreted. This apparent discrepancy is most likely due to the fact that palmitoylation stabilizes Tat association with the plasma membrane. Tat secretion and palmitoylation are thus concurrent mechanisms that apparently take place similarly in target and producer cells. This was intriguing since Tat secretion by infected cells is very active with ~2/3 of Tat being secreted[4], suggesting that Tat palmitoylation is prevented in infected cells.

**Cyclophilin A and FKBP12 are required for Tat palmitoylation.** In the search for cell proteins that might regulate Tat palmitoylation, we focused on prolylisomerases, i.e., proteins of the immunophilin family[34]. Indeed, although most proteins have <5% of Pro residues[35], Tat owns ~10 % of Pro residues. Four of them are located in the N-terminal region (in positions 6, 10, 14, and 18) and highly conserved among viral isolates[8], indicating that they are probably involved in Tat structure and/or biological activity.

We envisioned that prolylisomerases could be involved in Tat palmitoylation, as shown earlier for Ras[36]. We first used a pharmacological approach to examine the potential implication of the prolylisomerases CypA and FKBP12 in Tat palmitoylation. It can be seen in Fig. 6a that FK506 and rapamycin that inhibit FKBP12[36], and cyclosporin A (CSA) that inhibits CypA[37], strongly impaired Tat palmitoylation in PC12 cells. Accordingly, these inhibitors displace Tat from the membrane just as efficiently as 2-BP (Supplementary Fig. 12a). A cotransfected EGFP chimera of PH-PLCδ, an established ligand for PI(4,5)P$_2$, remained at the plasma membrane whatever the drug, showing that these inhibitors do not affect PI(4,5)P$_2$ availability (Supplementary Fig. 12a). These drugs strongly favored Tat secretion (by 2- to 6-fold; Fig. 6b), confirming that secretion and palmitoylation are

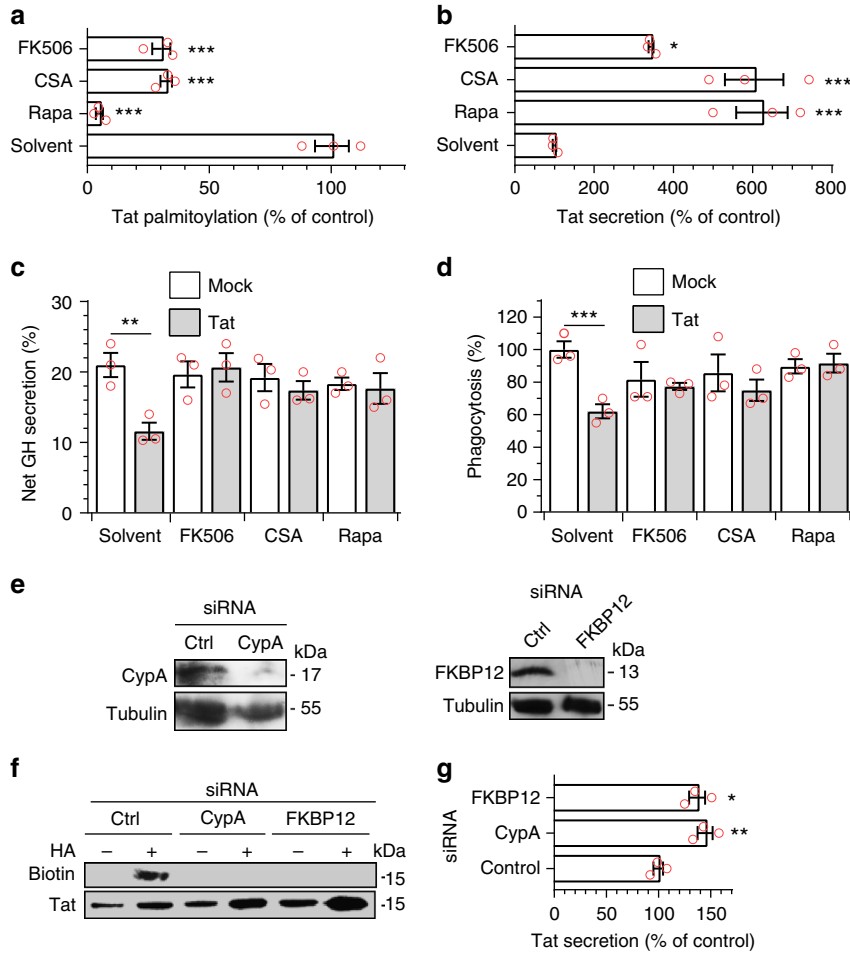

**Fig. 6** Cyclophilin A and FKBP12 are required for Tat palmitoylation and Tat-mediated inhibition of PI(4,5)P$_2$-dependent membrane traffic. **a** PC12 cells were transfected with Tat-FLAG before treatment for 6 h with 1 μM FK506, 2 μM CSA, 1 μM rapamycin or solvent. Palmitoylation was then assayed using acyl-biotin exchange before band quantification. Results are expressed as biotin/Tat intensity ratio (% of control). **b** Jurkat T-cells were transfected with Tat before adding drugs for 6 h and assaying Tat secretion by ELISA. **c** PC12 cells were transfected with GH before drug treatment for 6 h and GH secretion assay. **d** Human MDMs were treated with the drug for 5 h before assaying phagocytosis of IgG-coated beads. **e** PC12 cells were transfected with the indicated siRNA before lysis then anti-CypA or anti-FKBP12 then anti-αtubulin western blot. **f** PC12 cells were cotransfected with the indicated siRNA and Tat-FLAG before detecting Tat palmitoylation using acyl-biotin exchange. (HA, hydroxylamine). **g** Jurkat cells were cotransfected with Tat and the indicated siRNA (silencing efficiency is shown in Supplementary Fig. 12b) and Tat secretion was assayed after 48 h. Data are mean ± SEM, $n = 3$ independent experiments. The significance of differences with controls was assessed using ANOVA, one-way (**a**, **b**, **g**) or two-way (**c**, **d**) (***$p < 0.001$; **$p < 0.01$; *$p < 0.05$)

concurrent processes. These inhibitors also prevented Tat from affecting neurosecretion (Fig. 6c) and phagocytosis (Fig. 6d). Collectively, these pharmacological data indicated that the activity of both CypA and FKBP12 prolylisomerases is required for Tat palmitoylation. To confirm this point, we used siRNAs that efficiently inhibited the expression of CypA and FKBP12 (Fig. 6e). These siRNAs efficiently blocked Tat palmitoylation (Fig. 6f) confirming that both CypA and FKBP12 are required for Tat palmitoylation. Conversely, silencing CypA or FKBP12 in Jurkat cells (Supplementary Fig. 12b) increased Tat secretion by 40–60% (Fig. 6g). These results further confirmed that inhibition of Tat palmitoylation promotes its secretion by T cells and that CypA and FKBP12 are required for Tat palmitoylation.

**Tat interacts with CypA and DHHC-20.** We then examined whether Tat could interact with the proteins involved in its palmitoylation, i.e., CypA and DHHC-20. Following Tat-FLAG transfection into HEK 293 T cells, we indeed found that CypA and DHHC-20, but not DHHC-5, co-immunoprecipitated with Tat (Fig. 7a). These results confirm that DHHC-20 is involved in

Tat palmitoylation. CypA interacted less efficiently with non-palmitoylable Tat (Tat-C31S). To confirm these findings, we used GST pull down experiments. Both DHHC-5 and DHHC-20 could be recovered from cell extracts by GST-CypA (but not by GST), indicating that CypA interacts with these enzymes even in the absence of Tat (Fig. 7b). When Tat-transfected cells were used, Tat could be pulled-down by GST-CypA. Moreover, in the presence of Tat, ~4-fold more DHHC-20 was recovered by GST-CypA, indicating the existence of a Tat-CypA−DHHC-20 complex. The level of DHHC-5 pulled-down by GST-CypA was insensitive to the presence of Tat, confirming that DHHC-5 does not significantly interact with Tat. Altogether these results support the existence of a Tat palmitoylation complex containing Tat, CypA and DHHC-20.

**Tat interacts with FKBP12.** Although FKBP12 could be immunoprecipitated by Tat (Supplementary Fig. 13), GST-FKBP12 did not enable to pull-down Tat. We concluded that the interaction of Tat with FKBP12 is weaker than that with CypA. CypA was thus a better candidate to be used by the virus to modulate Tat

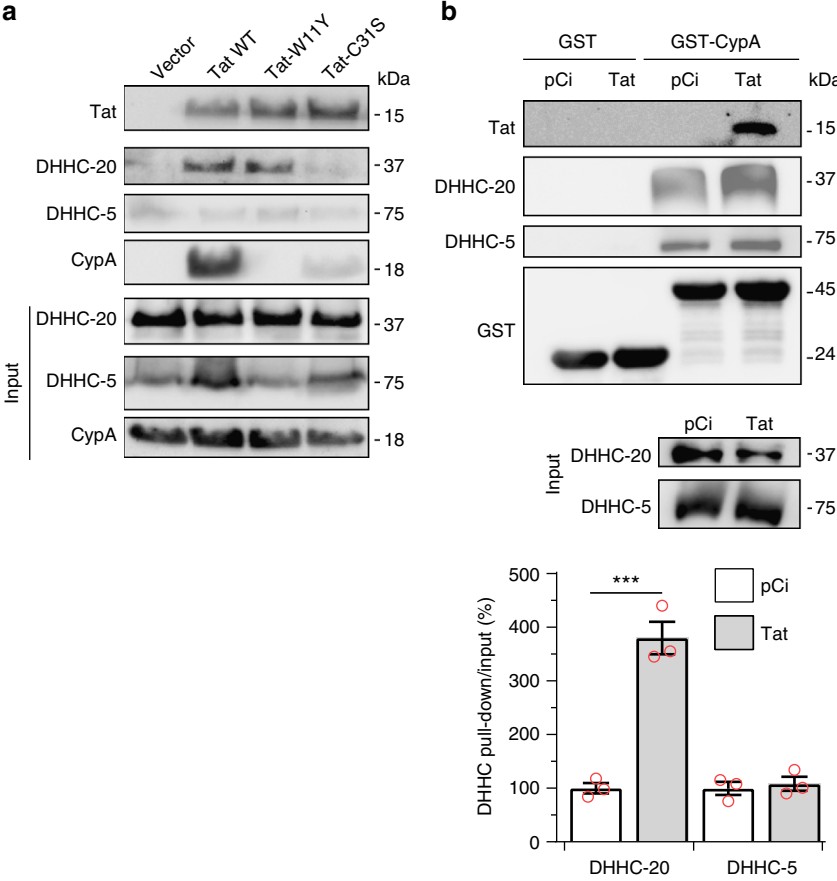

**Fig. 7** Tat interacts with CypA and DHHC-20. **a** HEK 293 T cells were transfected with an empty vector or Tat-FLAG (WT, 31 S, or 11Y). Cells were lysed 48 h after transfection before anti-FLAG immunoprecipitation and western blots against CypA, DHHC-5 and DHHC-20. **b** Cells were transfected with an empty (pCi) or Tat vector. GST or GST-CypA was added to cell extracts for GST pull-down before western blots. The graph shows the quantification of the DHHC pulled-down/input intensity ratio, setting the empty vector ratio to 100%. Representative data (mean ± SEM, $n = 3$ independent experiments) are shown.*** $p < 0.001$ (Student's $t$-test between Tat and vector)

palmitoylation. Moreover, while CypA is well known to be encapsidated by the virus (200 CypA /virion[38]), FKBP12 was found to be either not[39] or weakly encapsidated (25 FKBP12/virion)[40] and we thus focused on CypA.

**Gag inhibits Tat palmitoylation by exporting CypA.** HIV-1 Gag binds both PI(4,5)P$_2$[41] and CypA[38] through its matrix (MA) and capsid (CA) protein domain, respectively. Gag was therefore among HIV-1 proteins the best candidate to regulate Tat palmitoylation by modulating PI(4,5)P$_2$ and/or CypA availability for Tat. Consistently, when Tat and Gag were coexpressed in PC12 cells, Tat palmitoylation was abrogated (Fig. 8a). We first examined whether Gag could prevent Tat palmitoylation by inhibiting Tat recruitment to the plasma membrane, i.e., by displacing Tat from PI(4,5)P$_2$. As observed before[4], when primary CD4$^+$ T-cells were infected by HIV-1 (NL4.3), both Tat and Gag localize to the plasma membrane of infected T-cells (Fig. 8b). Hence, Gag does not prevent Tat palmitoylation by impairing Tat recruitment to the plasma membrane. Gag and Tat are both displaced to the cytosol in the presence of neomycin, a cationic antibiotic that tightly binds PI(4,5)P$_2$[42], thereby confirming the key role of this phosphoinositide in the localization of these HIV-1 proteins to the plasma membrane of infected T-cells (Fig. 8b).

Since Gag is unable to displace Tat from PI(4,5)P$_2$, we suspected that Gag prevented Tat palmitoylation by affecting CypA availability. Gag could either prevent Tat binding to CypA,

i.e., act by a titration effect and/or deplete cells from CypA due to encapsidation i.e., act by a depletion effect.

To discriminate between the titration and depletion effects, we prepared Gag mutants with the G89V mutation in CA to inhibit CypA binding[43] or the K22A and R27A mutations in MA to impair PI(4,5)P$_2$ binding[44]. The capacity of the CA-G89V mutation to prevent Gag-CypA interaction has been validated earlier[43]. We found that the MA-22/27A mutations prevented Gag recruitment to the T-cell plasma membrane (a PI(4,5)P$_2$-dependent process, Fig. 8b) while the CA-G89V mutation had no effect (Supplementary Fig. 14a). As expected, upon transient transfection the MA-22/27A mutations severely inhibited (by 40–50%) the formation of virus-like particles (VLPs) by T-cells while the CA-G89V mutation did not significantly affect the production of VLPs, although they are devoid of CypA (Supplementary Fig. 14b). When Gag-MA-22/27A or Gag-CA-G89V were cotransfected with Tat they did not significantly affect Tat palmitoylation (Fig. 8c). In the case of the CA-G89V mutant, this result indicates that CypA binding is needed for Gag to interfere with Tat palmitoylation, but this could be by a titration or a depletion effect. The fact that the MA-22/27A mutant that binds CypA but barely form VLPs (Supplementary Fig. 14b) does not affect palmitoylation thus indicates that VLP formation is required for Gag to shut off Tat palmitoylation. Altogether these data showed that Gag inhibits Tat palmitoylation by exporting CypA into VLPs and thus acts by depletion and not by titration.

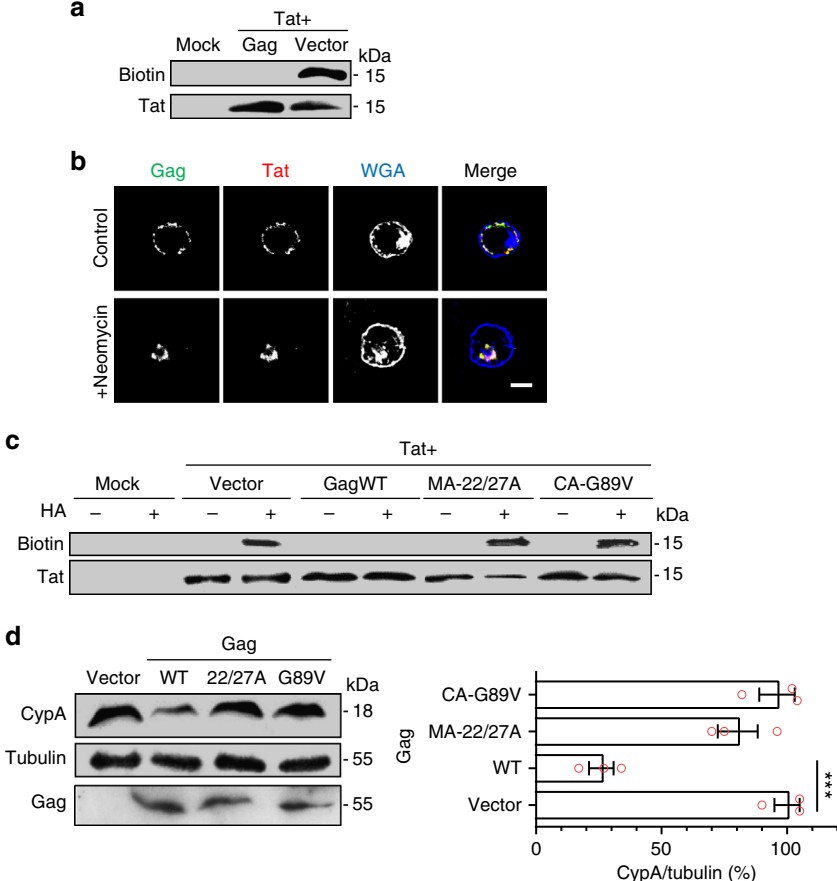

**Fig. 8** HIV-1 Gag inhibits Tat palmitoylation by exporting Cyclophilin A. **a** PC12 cells were cotransfected with Tat-Flag and Gag (1/1 ratio) or an empty vector as indicated. Cells were then labeled overnight with 17-ODYA before assaying Tat palmitoylation using click chemistry. **b** Human primary CD4[+] T-cells were infected with HIV-1 (NL4.3). After 24 h cells were processed for immunofluorescence detection of Tat and Gag, using fluorescent WGA to localize the plasma membrane and the TGN. When indicated, cells were pretreated with 5 mM neomycin for 1 h before fixation. Bar, 5 μm. **c** PC12 cells were cotransfected (1/1) with Tat-FLAG and a Gag mutant or an empty vector as indicated before assaying Tat palmitoylation using ABE. Gag-MA-22/27A and Gag-CA-G89V are mutated within their PI(4,5)P$_2$- and CypA-binding motif, respectively. **d** Jurkat T-cells were transfected with the indicated Gag mutant using nucleofection. Transfection efficiency was ~80%. After 24 h, cells were lysed for Western blots. Triplicate blots were used for the quantification. Data are mean ± SEM ($n = 3$) and were analyzed using one-way ANOVA (***$p < 0.001$)

**HIV-1 budding depletes cells in CypA.** CypA is an abundant protein present at 5–10 μM in the cytosol[37]. Hence, it was not obvious that the budding process could be sufficient to remove all CypA from the infected cell. We first examined this point using Western blotting. When Jurkat T-cells were transiently transfected with WT Gag, CypA level dropped by ~80% (Fig. 8d). Since transfection efficiency was also ~80%, this result indicated that WT Gag efficiently depletes T-cells in CypA. This was not the case when mutants such as Gag-CA-22/27 A (strongly affected in budding) or Gag-CA-G89V (unable to bind CypA) were used (Fig. 8d). This result was confirmed by immunofluorescence (Fig. 9a). Moreover, when cells were infected with pseudotyped HIV-1, infected cells showed after 24–30 h a strong depletion in their CypA content, while a virus unable to bind CypA (HIV-1-CA-G89V) did not affect CypA level (Fig. 9a). Altogether these data indicated that HIV-1 budding or transient transfection by Gag depletes cells of CypA.

**Tat is not palmitoylated in HIV-1 infected cells.** If our working model is correct, in infected T-cells, WT Tat secretion should be almost as efficient as that of Tat-C31S because Gag inhibits Tat palmitoylation by promoting CypA export. This was indeed the case (Fig. 9b), although Tat-C31S tends to be slightly more efficiently secreted. This is probably because Tat-C31S, contrary to

WT Tat, can be secreted before all CypA has been exported from the cell by virions. When a virus unable to bind CypA (HIV-1-CA-G89V) was used, Tat secretion was strongly inhibited (−50%), confirming that Gag inhibits Tat palmitoylation by exporting CypA. The negative control, Tat-W11Y that poorly binds PI(4,5)P$_2$ was not significantly secreted as observed before[4].

It was important to check that Tat is palmitoylated in HIV-1-CA-G89V infected cells to confirm our results in the context of infection, and to check that Gag, through its interaction with CypA, is the main viral regulator of Tat palmitoylation. To perform this experiment we had to circumvent a technical issue. Indeed, to follow Tat palmitoylation it is necessary to immuno-precipitate Tat and, since anti-Tat antibodies do not allow quantitative immunoprecipitation, Tat has to be tagged. This can only be done on its C-terminal side[45], and is not possible in the virus sequence because the three reading frames of Tat, Rev and Env overlap in this part of the sequence[46]. We thus chose to inactivate the *tat* gene in pNL4.3 and to cotransfect the resulting pNL4.3ΔTat or pNL4.3-CA-G89VΔTat with Tat-FLAG (WT or C31S) into Jurkat T-cells. This cotransfection allows for virus production (Supplementary Fig. 15). Tat palmitoylation was only observed when WT Tat, but not Tat-C31S, was cotransfected with pNL4.3-CA-G89VΔTat but not pNL4.3ΔTat (Fig. 9c) confirming, in infected cells, that Tat is indeed specifically palmitoylated on

Cys31 in a CypA-dependent manner. Moreover, the efficiency of WT Tat palmitoylation was essentially the same when cotransfected with pNL4.3-CA-G89VΔTat or with an empty vector. Since pNL4.3-CA-G89V encapsidates essentially as little FKBP12 as WT pNL4.3 (Supplementary Fig. 16), this result indicates that FKBP12 encapsidation is not involved in the inhibition of Tat palmitoylation observed in infected cells. More generally, this result shows that Gag-mediated CypA export is the main mechanism responsible for Tat palmitoylation inhibition in

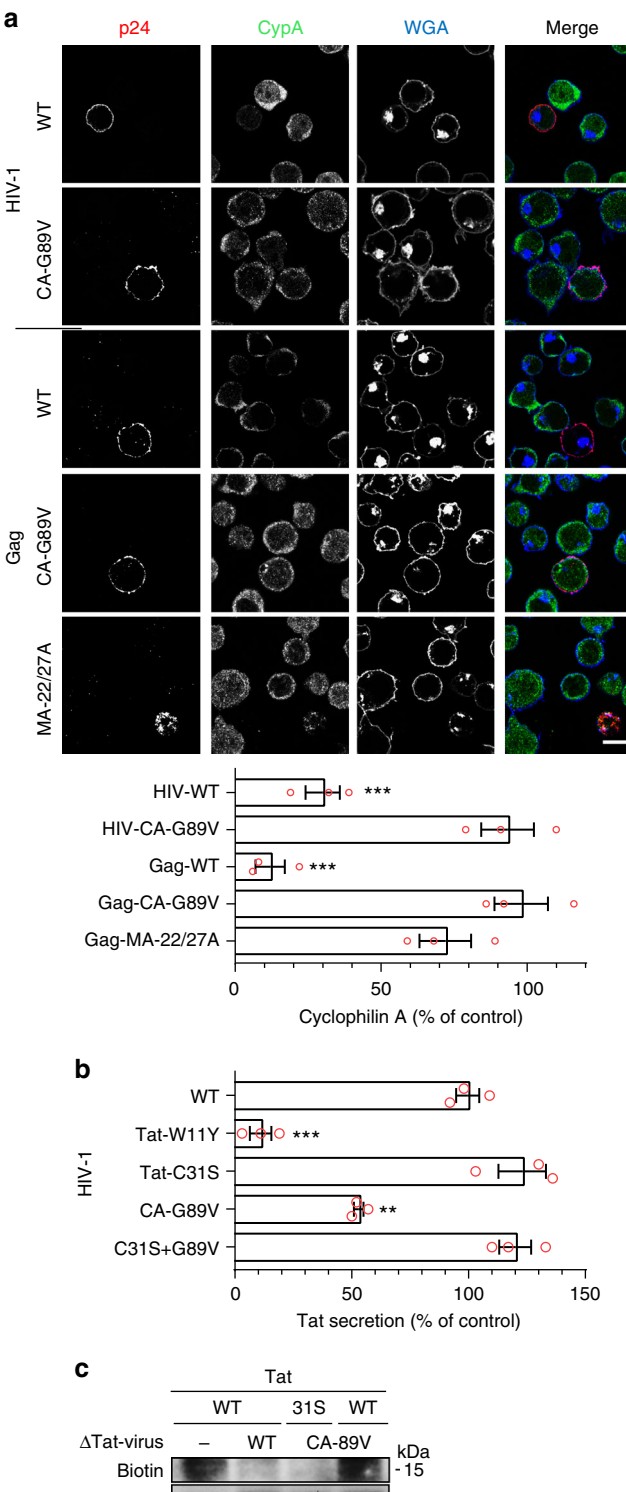

**a**

HIV-1 infected cells. Figure 10 is a schematic illustration of the results obtained in this study.

## Discussion

Most membrane traffic involving the plasma membrane uses PI $(4,5)P_2$-mediated protein recruitment at this level[47].

Here we showed that Tat palmitoylation is required for Tat to interfere with PI$(4,5)P_2$-dependent membrane traffic. Indeed, palmitoylation stabilizes the association of Tat with PI$(4,5)P_2$-membranes and prevents Tat secretion, while a non palmitoylable Tat (i.e., Tat-C31S) rapidly leaves the cell following PI$(4,5)P_2$ binding. Different cell types, such as T-cells, macrophages or neurosecretory cells can palmitoylate Tat. This result is consistent with the observation that Tat-palmitoylating enzyme, DHHC-20, is well expressed in most cell types[31] (Supplementary Fig. 7). Tat palmitoylation is a stable modification with a $t_{1/2}$ of 7.3 h, similar to the one of calnexin, a stably S-acylated protein[48]. Some bacterial virulence factors are also S-acylated by target cells. As it is the case for Tat, palmitoylation of these effectors is a stable modification that is required for durable association with the plasma membrane[49].

Tat is recognized as a major determinant of HIV-1 neuropathogenesis[18], and we identified Cys31 as the palmitoylated residue in Tat. Interestingly, this residue is replaced by a Ser in the Indian HIV-1 subtype C, a clade associated with a much lower neurotoxicity compared to other subtypes that owns a palmitoylable Tat[50]. Several studies identified Tat-Cys31 as a key residue responsible for the neurotoxicity of HIV-1 subtype B. Indeed, when applied onto primary neurons, Tat from HIV-1C showed several defects that include lower induction of chemokines, apoptosis induction, weaker mitochondrial membrane depolarization[51] and, when injected into mice, lower cognitive dysfunction[50] compared to Tat from HIV-1B. The dicysteine motif (C30–C31) is critical for these effects of exogenous Tat on human primary neurons and both Tat-C30S and Tat-C31S failed to elicit neurotoxicity[51]. To examine whether Tat palmitoylation could be involved in Tat deleterious effects on neuron activity we monitored whether exogenous Tat-C30S and Tat-C31S could be palmitoylated. Exogenous Tat-C31S is not palmitoylated since C31 is the palmitoylation site (Supplementary Fig. 4 for Jurkat and 17a for PC12 cells), but we also observed that exogenous Tat-C30S is not palmitoylated (Supplementary Fig. 17a). This might sound puzzling since transfected Tat-C30S is palmitoylated (Fig. 2), but recombinant Tat-C30S weakly binds to cells (57% of WT affinity, Supplementary Fig. 17b). Moreover, after overnight

**Fig. 9** Encapsidation-mediated depletion of cyclophilin A by HIV-1 inhibits Tat palmitoylation and enables strong Tat secretion. **a** Jurkat T-cells were infected with pseudotyped HIV-1 (NL4.3 WT or CA-G89V) or transfected with the indicated Gag mutant. After 24 h, cells were fixed for CypA and p24 staining by immunofluorescence. WGA was used to label the plasma membrane and the TGN. Representative median optical sections are shown (bar, 10 μm) together with the quantification from $20 < n < 30$ cells. Representative results of $n = 3$ experiments. **b** Primary human CD4+ T-cells were infected with a pseudotyped NL4.3 bearing either a mutation in Tat (TatW11Y, unable to bind PI$(4,5)P_2$ or Tat-C31S, non-palmitoylable) or/ and in CA (G89V, unable to bind CypA). Tat secretion was assayed by ELISA after 24 h. Results are expressed as percentage of Tat secretion by cells infected by the WT virus. Data (mean ± SEM, $n = 3$) were analyzed using one-way ANOVA (***$p < 0.001$; **$p < 0,01$). **c** Jurkat T cells were cotransfected (ratio 1/4) with Tat-FLAG or Tat-C31S-FLAG and an empty vector, pNL4.3ΔTat (ΔTat-virus WT) or pNL4.3CA-G89V-ΔTat (ΔTat-virus CA-G89V) as indicated. Palmitoylation was then assayed using ABE before anti-Tat western blot. Representative experiment of $n = 3$

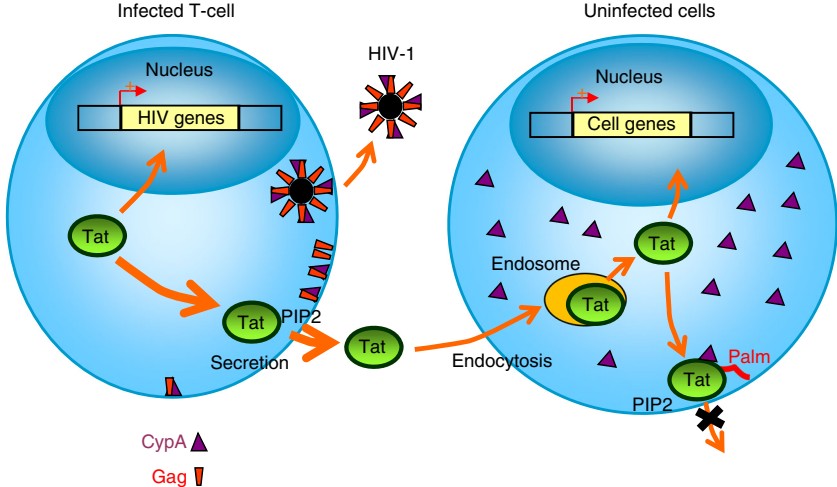

**Fig. 10** Schematic representation of the results. Neosynthesized Tat enters the nucleus to ensure efficient transcription of viral genes, but most (~60%) of Tat is secreted by infected cells using an unconventional mechanism based on the recruitment of Tat by PI(4,5)P$_2$ at the plasma membrane. Gag multimerization drives virus assembly at the plasma membrane. Because Gag binds CypA, virus budding depletes cells of CypA thereby inhibiting Tat palmitoylation specifically and only in infected cells. Secreted Tat can enter target cells (such as macrophages, neurons, myocytes, etc) by endocytosis before translocation to the cytosol from endosomes. Tat can then go to the nucleus to affect the transcription of cellular genes, but most of it is recruited by PI(4,5)P$_2$ at the plasma membrane, before CypA-dependent palmitoylation. Tat palmitoylation inhibits its secretion and enables cumulative effects of minute Tat doses that will efficiently impair PI(4,5)P$_2$-dependent cell activities such as, depending on cell types, phagocytosis, neurosecretion and ion transport (not depicted)

labeling exogenous Tat-C30S is not detectable in the cytosol, even on overexposed western (supplementary Fig. 17a). Recombinant Tat-C30S apparently suffers from a conformational problem. Indeed, it does not firmly bind to the heparin-agarose column used during Tat purification, indicating that Tat basic domain is not properly exposed in this mutant. The fact that both exogenous Tat-C30S and Tat-C31S failed to be palmitoylated indicates that the requirement for Tat dicysteine motif to elicit neurotoxicity on primary neurons might be due to the need for Tat to be palmitoylated and thereby become resident on PI(4,5)P$_2$ to trigger neuronal toxicity. It is indeed well established that PI(4,5)P$_2$ is a key lipid for neuron biology[52]. Since we found that Tat inhibits both neurosecretion[21] and key potassium ion channels[22], it is thus tempting to speculate that the low neuropathogenesis of the Indian clade C involves the inability of its Tat to become palmitoylated. Alternatively, since the dicysteine motif of Tat is also present in several chemokines[53], the absence of chemokine-like activity of Indian clade C Tat[54] might also be implicated in its lower neurotoxicity.

We found that both CypA and FKBP12 are required for Tat palmitoylation. In the case of Tat, immunophilins are required for palmitoylation, while FKBP12 was found to promote H-Ras depalmitoylation[36]. These chaperones thus regulate palmitoylation in a substrate-dependent manner. As Tat transmembrane transport requires unfolding[55], it is possible that prolylisomerases inhibit Tat secretion by favoring or inducing Tat folding, in such a way that DHHC-20 has easy access to Tat-Cys31. Accordingly, we found that both CypA and FKBP12 interact with Tat, although Tat- FKBP12 interaction seems to be weaker and not involved in Tat palmitoylation regulation by HIV-1. We also obtained evidence for a Tat-CypA-DHHC-20 complex. Both the organization and the stoichiometry of the complex components are unclear. Indeed, CypA has a single catalytic domain that should not allow it to have more than one partner at the same time.

Why does Tat palmitoylation require both FKBP12 and CypA? The specificity of immunophilins is known to be rather low[37]. There is nevertheless some substrate specificity that is, at least for CypA and FKBP12, largely due to the residue before the Pro. Hence, CypA prefers small residues, with the GP motif being the most efficient[56]. This motif is present in HIV-1 capsid protein (CA-Gly89-Pro90)[38]. FKBP12 prefers large hydrophobic residues such as Leu or Phe before the Pro[57]. Hence, there is minimum specificity overlap between CypA and FKBP12. It was therefore not surprising that both chaperones were needed for Tat palmitoylation. Regarding the possible target prolines for CypA in the palmitoylation complex, both Tat (Gly83-Pro84) and DHHC-20 (Gly253-Pro254) own a GP motif. Whether CypA binding to these Pro residues is required for Tat palmitoylation remains to be established. More generally, the Pro residues that need to interact with CypA or FKBP12 to insure robust Tat palmitoylation by the palmitoylation complex await identification.

During HIV-1 assembly, CypA directly binds CA-Pro90 and is thereby encapsidated with an efficiency of ~ 200 copies of CypA/virion[38]. We found that CypA encapsidation by HIV-1 depletes T-cells of CypA. This finding is consistent with known concentration of CypA in the cytosol, i.e., ~0.25% of cytosolic proteins[37]. This corresponds to ~$10^7$ molecules/cell and we accordingly measured using recombinant human CypA as standard and a semi quantitative Western blot assay ~$6.10^6$ molecules/Jurkat T-cell. Since HIV-1 maximum production rate is approximately $5 \times 10^4$ virions/24 h[58], this means that HIV-1 can daily exports ~$10^7$ CypA molecules, i.e., approximately twice the cellular stock of CypA. The impact of CypA removal on cell viability is likely moderate. Indeed, a CypA-deficient Jurkat cell line is only weakly affected in growth rate[24], and a CypA-deficient mice is fully viable[59].

Gag-mediated export of CypA strongly favors efficient Tat secretion that will be maximum at late stages of the viral cycle when budding events will have exported all cellular CypA. Hence, this regulatory system enables the virus to keep Tat intracellular during the early stages of the viral cycle when Tat is needed to boost viral transcription while permitting strong Tat secretion at late stages when Tat is more abundant. In target cells, i.e., non-infected cells in which extracellular Tat enters by endocytosis before translocation to the cytosol, Gag is absent and Tat is

efficiently palmitoylated, locking it on PI(4,5)P$_2$, reducing its secretion and enabling cumulative effects of Tat doses. Hence, minute Tat concentration can efficiently interfere with PI(4,5)P$_2$-dependent processes. This PI(4,5)P$_2$-targeting toxin is likely involved in the development of (i) opportunistic infections, due to its capacity to inhibit phagocytosis[20], (ii) HIV-1 associated neurological disorders due to Tat ability to impair neurotransmitter secretion[21] and (iii) cardiac disorders linked to Tat inhibition of key cardiac ion channels[22]. These defects have been documented in HIV-1 infected patients[20–22]. The effects of Tat on other PI(4,5)P$_2$-dependent machineries remains to be demonstrated but available data indicate that palmitoylation enables Tat to act as a wide range PI(4,5)P$_2$-toxin affecting most of the numerous processes in which this phosphoinositide is involved.

## Methods

**Materials**. Antibodies were obtained as follows. Polyclonal rabbit anti-Tat antibodies (a kind gift of Dr E. Loret, Marseille), monoclonal anti-Tat (SantaCruz sc-65912 or 65913), goat anti-p24 antibodies (Serotec 4999-9007), anti-CypA (sc134910), anti-FKBP12 (Pierce PA1-026A), anti-MAP2 (a generous gift of Pr. Klosen, INCI Strasbourg), anti-DHHC-5 (Sigma HPA014670), anti-DHHC-20 (Sigma SAB 4501054), anti-flotillin-2 (sc-28320) and anti-αtubulin (Sigma T5168) were obtained as indicated. The specificity of anti-CypA antibodies was checked using Jurkat cells deficient for CypA[24]. Anti-FLAG M2 antibodies and anti-FLAG M2 Magnetic beads were from Sigma. Kc57-RD1 anti-p24 antibody for FACS analysis was from Beckman-Coulter. Anti-24 antibodies for p24 ELISA (ARP410 and ARP454) were from the NIBSC/CFAR bank (England). Secondary antibodies were all from Jackson Immunoresearch. Antibodies were at ~1 μg/ml for immunofluorescence and ~30 ng/ml for Western blots. Wheat Germ Agglutinin (WGA), fluorescent phalloidins, Opti-MEM, Lipofectamine 2000 and biotin-azide were from Life technologies. Cyclosporin A (CSA), rapamycin, and FK506 were from Merck-Millipore. 17-ODYA was from Cayman Chemical, EZ-link®HPDP-Biotin from Thermo Scientific and PEImax from Tebu-bio. Protease inhibitor cocktail (Complete) was from Roche. Tris(2-carboxyethyl)phosphine (TCEP), tris [(1-benzyl-1H-1,2,3-triazol-4-yl)methyl]amine (TBTA), ExtrAvidin-conjugated horseradish peroxidase, palmitoyl chloride, most chemicals, oligos for qRT-PCR and Mission siRNAs were from Sigma. Fetal bovine serum (Life technologies) was delipidated by extensive dialysis against PBS at 4 °C. The pBI-Tat expression vector has been described before[4]. A FLAG epitope (Asp-Tyr-Lys-Asp-Asp-Asp-Asp-Lys) was attached to Tat C-terminus by PCR. Tat Cys to Ser mutants were generated using Quickchange (Agilent Technologies) and mutants were entirely sequenced. Mouse DHHC GFP-expression vectors and human Myc-tagged DHHC plasmids were from Christophe Lamaze and Laurence Abrami, respectively. To prepare solutions containing fatty acids (palmitate or 2-BP), they were first added (1 mM in DMSO) to 5% defatted BSA then to delipidated serum and incubated 3 min at 37 °C before adding to cells.

**Recombinant proteins**. Recombinant Tat (WT, C31S, W11Y or His$_6$) was produced and purified from transfected *E. coli*, as described[10]. Tat devoid of all Cys except C31 (TatC31O) was synthesized by Proteogenix (Schiltigheim, France). Its identity and purity (>96%) was verified by mass spectrometry (Supplementary Fig. 9). It was palmitoylated using palmitoyl chloride (Sigma) essentially as indicated[60]. Briefly, 1.65 mg of Tat-C31O (170 nmoles) was dissolved in 100 μl trifluoro acetic acid (TFA) and half of it was reacted for 10 min at RT with (or without for the control protein) a 10-molar excess of palmitoyl chloride. The reaction was quenched with ethanol before injection of Tat-C31O and Tat-C31O-Palm on a C4 HPLC column and elution with a gradient of acetonitrile in water. TFA (0.1%) was added to solvents[10]. The purified proteins were then aliquoted, dried under vacuum (Eppendorf concentrator plus) and stored at −80 °C. MALDI-TOF analysis was performed by the IRCM facility (Montpellier, France) using sinapinic acid (10 mg/ml in 50% acetonitrile/TFA 0.1%) as a matrix and a 4800 Plus MALDI-TOF/TOF Proteomics Analyser (ABSciex).

GST-FKBP12 and GST-CypA (kindly provided by Mark Phillips and Phillipe Gallay, respectively) were produced in *E. coli* BL21, purified on glutathione-agarose as described[61] and stored at −80 °C. These constructions have been validated in previous studies[36,62]. Recombinant HIV-1 capsid protein (p24) was purified from transfected *E.coli*, as described[63].

**Cells and transfections**. All cell lines were obtained from the ATCC and cultivated following their recommendations. Cell lines were routinely screened (every 1–2 months) for mycoplasma contamination using MycoAlert (Lonza). PC12 cells (ATCC CRL-1721) were seeded at 70–80% confluence in a T75 flask the day before transfection. One hour before transfection the medium was replaced by Opti-MEM medium (Life Technologies). Cells were then transfected with DNA or cotransfected with DNA and siRNAs using Lipofectamine 2000 or 3000 and the protocol described by the manufacturer. After 5 h, the transfection medium was replaced by

complete medium. When indicated cells were treated for 5 h with 100 μM 2-BP, 2 μM CSA, 1 μM rapamycin or 1 μM FK506.

Jurkat cells (ATCC TIB-152) were transfected by electroporation[10], or using nucleofection and the protocol of the manufacturer (Amaxa). HEK 293 T cells (ATCC CRL-11268) were transfected using PEImax as described[64].

For the preparation of human monocytes and T-cells, human blood was obtained from the local blood bank (Etablissement Français du Sang, Montpellier) according to the French rules and the agreement 21/PLER/MTP/CNR11/2013-049 between EFS and IRIM. PBMCs were isolated by separation on Ficoll-Hypaque (Eurobio). Monocytes and CD4$^+$ T-cells were then purified using CD14 microbeads (Miltenyi Biotec) and a CD4 easysep negative selection kit (Stemcell), respectively. Monocytes were differentiated to macrophages by cultivation for 6–8 days in the presence of 50 ng/ml Macrophage colony-stimulating factor (Immunotools), before phagocytosis assays. CD4 + T-cells were activated using phytohemagglutinin (1 μg/ml) for 24 h then interleukin-2 (50 U/ml) for 6 days before nucleofection (as recommended by the manufacturer) or infection by HIV-1. Chromaffin cells were isolated from fresh bovine adrenal glands and maintained in primary culture as described previously[21].

New-born wild type C57BL/6 mice pups were obtained from the INCI animal house that purchases adult mice from Charles River Laboratories (Saint-Germain-Nuelles, France). The breading was carried out in house under controlled conditions (Authorization No. A67-2018-38). For neuronal cultures new born pups (P0) were sacrificed respecting regular ethical statement—Authorization No. AL/01/26/11/12—according to the French Ministry of Research and ethic commission (CREMEAS). Neurons were transfected with a Tat (WT or C31S) vector on the day of culture by the Neon electroporation system (Invitrogen) using 1 μg of DNA for 1 million cells in 100 μl of electroporation buffer. Neurons (30 μl) were then plated onto coverslips in 24-well plates and were fixed after 24 h either directly or following 4 h incubation with 50 μM palmitate or 2-BP. Similar data were obtained following neuron transfection with Lipofectamine 2000.

**Transactivation assays**. PC12 cells were cotransfected with a Tat vector, a vector expressing a Firefly gene under the control of an LTR promoter, a plasmid with a Renilla gene under a CMV promoter (both described before[4]) and a human cyclin T1 vector (provided by Dr R. Kiernan, Montpellier). The latter was required because Cyclin T1 is needed for Tat transactivation and rodent cyclin T1 does not support transactivation[65]. When Extracellular Tat was tested, cells were not transfected with Tat but treated with 200 nM recombinant Tat for 24 h. Luciferase activities were then assayed and transactivation is expressed as the Firefly/Renilla ratio (%)[10].

**Infections**. The quickchange method was used to introduce Tat-C31S, ΔTat (M1T, D5G, L8STOP), and CA-G89V mutations in the NL4-3 virus that owns a 101 residues Tat protein[4]. The amplified regions were entirely sequenced. To produce pseudotyped virions HEK 293 T cells were transfected with pNL4.3 and pCMV-VSV-G vector (Addgene) using a 5:1 ratio. After 48 h the supernatant was harvested, filtered onto 0.22 μm filters and ultracentrifugated on 20% sucrose at 125 000 × g for 2 h at 4 °C[66]. Viruses were stored at −80 °C in aliquots. Virus titers were determined using Jurkat cells stained with kc57-RD1 and analyzed by FACS. Infections were performed using MOI of 0.2-1. For palmitoylation experiments Jurkat T cells were transfected with pNL4.3 ΔTat and pBi-Tat-FLAG using a 4/1 ratio.

**p24 ELISA**. The protocol 3 of Aalto Bio Reagents Ltd (Dublin, Ireland) was followed with minor modifications. White 96-plates were coated with a mixture of three sheep anti-Gag antibodies (ARP410) at 5 μg/ml. The plate was then washed with TBS before saturation with BSA (3% in TBS), then incubated with samples and p24 standards (both diluted in TBS containing 0.1% Empigen BB (Sigma)). A biotinylated monoclonal anti-p24 antibody (ARP454) was then added, and finally extravidin-peroxidase. Peroxidase activity was assayed using Luminata forte and a Berthold luminometer.

**Click chemistry**. The original method was used[25] with slight modifications. For transfected Tat-FLAG labeling with 17-ODYA, 4 h after transfection the medium was changed to medium with delipidated serum (delipidated medium) to starve cells. Cells were then washed with PBS before overnight incubation in delipidated medium containing 100 μM of 17-ODYA previously complexed to 10% fatty acid-free BSA. Cells were then washed once in ice-cold PBS, harvested using a cell scraper, collected by centrifugation (400 g, 5 min, 4 °C) and lysed in lysis buffer [150 mM NaCl, 50 mM citric acid, 1% Triton X-100, protease inhibitors, 10 mM N-ethylmaleimide (NEM), 0.5% 3-[(3-Cholamidopropyl)dimethylammonio]-1-propanesulfonate (CHAPS)]. The lysate was clarified by centrifugation (20000 × g, for 20 min at 4 °C). 10 μL of anti-FLAG magnetic beads were then added to clarified lysates before incubation at 4 °C for 1 h with end-over-end rotation. Magnetic beads were washed three times in lysis buffer then once in TC buffer (150 mM NaCl, 50 mM Tris-HCl, 1% TX-100, pH 7.4).

When using recombinant Tat-His$_6$, this protocol was modified as follows. 50 nM Tat-His$_6$ was added together with 100 μM 17-ODYA and Tat-His$_6$ was purified from cell lysates using Ni-NTA agarose beads (Qiagen). Beads were finally

resuspended in Click solution (100 μM biotin azide, 1 mM TCEP, 100 μM TBTA,1 mM CuSO₄ in TC). All these solutions were freshly made. The reaction mixture was incubated at room temperature for 1 h with end-over-end rotation. Beads were then washed twice in TC buffer and resuspended in 30 μl of SDS-PAGE sample buffer containing 20 mM DTT before analysis on 16% acrylamide Tris-Tricine gels. After SDS–PAGE and transfer to nitrocellulose membrane, membranes were blocked with 4% BSA. Detection of biotinylated Tat was performed using an ExtrAvidin-conjugated peroxidase (Sigma; 1:5000 in 4% BSA), before detecting Tat using a monoclonal anti-Tat antibody (1:2000) and an HRP-conjugated goat anti-mouse IgG (1:3000). Uncropped blots are presented in Supplementary Fig. 18.

**Acyl-biotin exchange.** These experiments were performed essentially as described earlier[26]. One day after transfection with Tat-FLAG, cells were harvested using a cell scraper then lysed with lysis buffer. Anti-FLAG magnetic beads (10 μl) were then added to cleared cell lysates before incubation at 4 °C for 1 h with end-over-end rotation. Magnetic beads were washed three times in lysis buffer then once in TC buffer. Samples were reduced with 10 mM TCEP in TC buffer for 30 min at room temperature, then alkylated with 50 mM NEM for 2.5 h at room temperature with end-over-end rotation. Excess NEM was removed with four washes in TC buffer. During the last wash, samples were divided equally in two. To one half (HA⁺) freshly prepared hydroxylamine solution (1 M hydroxylamine, 1% TritonX-100, 0.5 mM EZ-link®HPDP-Biotin, pH 7.4) was added, while the second half (HA⁻) received the same solution but containing 50 mM Tris-HCl, pH 7.4 instead of hydroxylamine. Samples were incubated at room temperature for 1 h with end-over-end rotation then washed three times in TC buffer. Beads were finally resuspended in 30 μl of non-reducing SDS-PAGE sample buffer before analysis on 16% acrylamide Tris-Tricine gels. Biotinylated Tat was detected, as described above.

**Acyl-RAC.** These experiments were performed as described[32], except that 50 mM NEM was used instead of S-methyl methanethiosulfonate to block free SH groups. Briefly, cells were disrupted by passing ten times through a 26 G needle. The post nuclear supernatant was treated with NEM, proteins were precipitated and washed using cold acetone before solubilization using 35 mM SDS. They were then treated with hydroxylamine (HA) or Tris as a control in the presence of Thiopropyl Sepharose beads (GE Healthcare) to trap proteins exposing free SH groups. Unbound proteins were collected and bound proteins were eluted by two sequential extractions with SDS-PAGE reducing sample buffer. Acylation was quantified as Bound (HA)/[Bound (HA) + Unbound (HA)]−Bound (Tris)/[Bound (Tris) + Unbound (Tris)] and expressed as a percentage.

**GST pull downs.** HEK 293 T cells were lysed in lysis buffer before centrifugation. GST proteins on glutathione sepharose (10 μl) were then added to the lysate (50 μl) diluted to 500 μL with lysis buffer. Following 1 h at 4 °C with end-over-end rotation, the resin was washed three times with lysis buffer then resuspended in reducing SDS/PAGE sample buffer for analysis.

**Immunofluorescence.** PC12 cells were plated onto alcian-blue coated coverslips the day before fixation while primary T-cells were allowed to adhere to alcian-blue coverslips for 3 min at room temperature. Cells were fixed with 3.7% paraformaldehyde before permeabilization with 0.1% saponin. Tat, Gag or GM130 were then detected by indirect immunofluorescence. WGA-Cy5 (5 μg/ml) was used to label glycoproteins on the plasma membrane and in the TGN, and fluorescent phalloidins (0.1 μg/ml) to detect F-actin[4]. For endocytosis assays, PC12 cells were labeled for 30 min on ice with 100 nM Tat before washing and chasing for the indicated times before fixation and F-actin staining using fluorescent phalloidin. Tat membrane localization was assessed by quantifying Tat/F-actin colocalization using confocal images from 50 < n < 100 cells and Mander's coefficient calculation[20].

**qRT-PCR.** Total RNA was extracted using Trizol before reverse transcription using SuperScript™ III Reverse Transcriptase and oligodT primers. cDNAs were then amplified by real time PCR in triplicates using the LightCycler 480 SYBR Green I Master mix on LightCycler 480 thermo cycler (Roche Applied Science). Primers for DHHCs were listed in Supplementary Table 2 were chosen using Primer Blast (http://www.ncbi.nlm.nih.gov/tools/primer-blast/) and quantification was performed relative to the expression of GAPDH using the delta delta Ct method[67].

**Phagocytosis assays.** Phagocytosis assays were performed as described[20]. Briefly, phagocytic targets were 3 μm latex beads (LB30, Sigma) coated with human IgGs. MDMs were pretreated for 3 h with 5 nM Tat (WT or C31S), 100 μM 2-BP, 2 μM CSA, 1 μM rapamycin or 1 μM FK506. Phagocytosis was synchronized by bead centrifugation on the cells for 1 min at 100 × g before incubating the plates at 37 °C for 7–10 min. Plates were then placed on ice and washed before labeling extra-cellular beads using fluorescent Cy5-labeled anti-human IgGs donkey Fab' for 20 min at 4 °C, fixation and bead counting using a fluorescent microscope. More than 100 cells were scored for each condition. The mean number of phagocytosed targets was calculated, and results are expressed as percentage of control cells (mean ± SEM; n = 3 experiments performed using MDMs from different donors)[20].

The infected T-cells/macrophage coculture setup was detailed before[20]. Briefly, CD4 + T cells were purified, stimulated for 6 days then infected (MOI 0.5) with a T-tropic (NL4.3) virus, bearing either WT Tat or Tat-W11Y. After 1 day, T cells were washed then added to Transwells into wells containing autologuous MDMs (10 T-cells per MDMs). Cells were cocultured for 8 days before assaying FcγR-mediated phagocytosis. At this time, 20–35% of T cells were infected while macrophages remained uninfected.

**Neurosecretion and Tat secretion assays.** To monitor the effect of Tat on neurosecretion, PC12 or chromaffin cells were transfected with GH[21]. 24 h after transfection, cells were treated with 20 nM Tat for 5 h then incubated for 10 min in calcium-free Locke's solution (basal release) or stimulated for 10 min with a depolarizing concentration of K⁺ (K59; Locke's solution containing 59 mm KCl and 85 mm NaCl). Supernatants were collected and cells were lysed. The amounts of secreted and intracellular GH were then assayed using ELISA (Roche Applied Science).

Tat secretion by Jurkat, primary T-cells or PC12 was assayed as described[4,33].

**Surface plasmon resonance and microcalorimetry.** Surface plasmon resonance (Biacore) and ITC experiments were performed using PI(4,5)P₂ liposomes as described[4].

**Statistical analyses.** Sample size was chosen to insure data significance following statistical analyses. Data are presented as mean ± SEM and individual data points are indicated on bar graphs. Two-sided Student's t-tests were used to compare two conditions, one way ANOVA with Dunnet post-hoc test when more than two conditions were compared, and two way ANOVA with Bonferroni post-hoc test when there were two independent variables (mutations and drugs for instance). Data analyses were performed using GraphPad Prism 7. p-Values on the figures are indicated as *p < 0.05, **p < 0.01, ***p < 0.001 and n.s. for p > 0.05.

**Protein preparation for in-gel digestion.** For proteomic analysis, PC12 cells were transfected with Tat-FLAG before immunoprecipitation and SDS-PAGE. Bands around 15 kDa were excised, gel pieces were washed with 25 mM NH₄HCO₃, dehydrated with acetonitrile before reduction with 10 mM DTT in 25 mM NH₄HCO₃ (1 h at 57 °C) and alkylation with 55 mM iodoacetamide. Gel pieces were resuspended in 2 volumes of freshly diluted trypsin (12.5 ng/μL) and incubated overnight at 37 °C. Digested peptides were then extracted from the gel using 34.9% H₂O, 65% acetonitrile and 0.1% HCOOH before drying.

**Nano HPLC and MS and MS/MS.** The analysis was performed on an Ultra-Performance-LC (Waters). Samples were separated on a C18 column (Waters Corp.) using 0.1% formic acid in water and 0.1% formic acid in acetonitrile. MS and MS/MS analyzes were performed on SYNAPT™ (Waters) equipped with a Z-spray ion source and a lock mass system. The capillary voltage was set at 3.5 KV and the cone voltage at 35 V. Online calibration was performed with Glu-fibrino-peptide B as the lock-mass. The 3 most abundant peptides (intensity threshold 60 counts/s), preferably doubly and triply charged ions, were selected on each MS spectrum for further isolation and CID fragmentation with 2 energies set. Fragmentation was performed using argon as the collision gas. The complete system was fully controlled by MassLynx 4.1 (Waters). Raw data were processed with ProteinLynx Browser 2.5 (Waters). Normal background subtraction type was used for both MS and MS/MS with 5% threshold and polynomial correction of order 5, and deisotoping was performed. The trypsin peak lists were searched using Mascot (Matrix science). During database search, four variable modifications (oxidation of Methionine (+16 Da), carbamidomethylation of Cysteine (+57 Da), acetylation on Lysine (+42 Da) and palmitoylation on Cysteine (+238 Da)) were considered. The search window was set to 25 ppm for precursor ions and 0.07 Da for fragment ions. Filtering criteria based on probability-based scoring of the identified peptides were taken into account: 2 peptides were needed to identify a protein and each peptide with a Mascot Ion score >25 is validated. The false discovery rate (FDR) was calculated to be <1 % based on the number of decoy hits.

**Data availability.** All data are available from the authors upon request.

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

## Acknowledgements
We are indebted to Christophe Lamaze, Cedric Blouin and Laurence Abrami for DHHC clones, Erwann Loret for anti-Tat antibodies, Corinne Lionne, Laurent Chaloin, and Gudrun Aldrian for help with Tat HPLC-purification, Simon Lachambre and Fanny Martini for pioneer phagocytosis experiments, Jean-François Guichou for expert assistance during ITC experiments, Nicole Bec and Sarah Cianferani for MALDI-TOF/TOF analysis and Sara Salinas for providing primary neurons. This work was funded by ANRS, Sidaction and the FRM (Equipe FRM 20161136701) to B.B. and by ANR and FRM to N.V.

## Author contributions
C.C., V.T., and H.Y. performed palmitoylation experiments. C.C. prepared all Tat and Gag mutants. P.T., A.T., and N.V. performed neurosecretion experiments. P.T. carried out experiments on primary neurons. S.D., M.S., P.T., and V.T. performed Tat secretion experiments. C.C., V.T., and M.S. performed proteins interaction studies. C.M. prepared pNL4.3 mutants. C.C. and A.G. performed RT-PCR experiments. M.P. and J.-M.S. supervised Biacore and MS/MS experiments, respectively. J.-M.M. provided intellectual input. B.B. designed the study, performed confocal microscopy and wrote the paper.

## Additional information

**Competing interests:** The authors declare no competing interests.

