## [Peer Review File · Nature Communications]

Reviewers' comments:

Reviewer #1 (Remarks to the Author):

In the study by Chopard et al., the authors demonstrate that the HIV protein Tat palmitoylation increases its affinity for PI(4,5)P₂ thereby enhancing its accumulation in the membrane of uninfected cells, while infected cells suppress Tat palmitoylation through Gag-dependent inhibition of cyclophilin A, DHHC20, and palmitoylation. Overall, this is an important manuscript that presents significant new information on the molecular biology of HIV.

Specifically, this work identifies a cyclophilin A-dependent mechanism of Tat palmitoylation. Palmitoylated Tat PI(4,5)P₂ binding locks it into the membrane (7.3 h = t_{1/2}) hindering viral budding and exocytosis, as well as phagocytosis in macrophages. This mechanism, problematic for infected cells, is suppressed by Gag, which sequesters peptidyl prolyl isomerase A/cyclophilin A and DHHC20. The study is well designed, and well explained throughout, and is a logical extension of previous work by this group showing Tat binds to the phosphorylated inositol of PI(4,5)P₂ and embeds in the membrane via the Trp residue. The authors use complementary gain-of-function, loss-of-function experiments to systematically explore the role of HIV-dependent alterations in cyclophilin A and DHHCs in Tat trafficking/membrane accumulation. The findings provide answers to previously unresolved questions regarding Tat. The data address why extremely low levels of Tat can display cytotoxicity in bystander but not infected cells, and further explains Tat's numerous actions independent of its role as a transcription factor—especially regarding its role as a cell penetrating protein. The study also provides insight into why Tat derived from clade C virus, which lacks Cys31—a critical palmitoylation site for PI(4,5)P₂ binding, is less neurotoxic than Tat derived from other clades in which Cys31 is highly conserved. Minor comments and corrections are outlined below:

The authors confirm aspects of their findings in Jurkat, HEK293, and PC12 cell lines, which is dependent on DHHC20, using primary PBMCs and chromaffin cells, but do not examine the role of Tat palmitoylation in primary neurons. PC12 cells can be considered preneurons and poor surrogates of primary neurons. Is Tat Palmitoylated in primary neurons (as shown in Supplementary Table 1) and in sufficient amounts to cause neurotoxicity?

Page 8, lines 140-143; the sentence talking about biphasic membrane localization using Tat/F-actin colocalization is confusing and needs rewriting.

Page 9, lines 193-195; the sentence referring to Tat ownership of Pro residues is unclear and needs rewriting.

Although not incorrect, rarely used terms such as *in cellulo* might be replaced with "in living cells".

Reviewer #2 (Remarks to the Author):

Chopard and colleagues seek to determine how Tat secretion from HIV infected cells versus Tat retention in target, uninfected cells is regulated. They show that Tat is palmitoylated on C31 in uninfected cells but is not palmitoylated in infected cells. Palmitoylation of Tat is mediated by DHHC20 and requires Cyclophilin A, which is depleted by encapsidation in infected cells.

The data presented are sound for the most part (although see specific points raised below). However, greater depth of insight into (1) the pathogenic relevance and (2) the directness of the mechanism of CypA-dependent palmitoylation of Tat would make this study more appropriate for publication in Nature Communications. These points are likely best addressed in the form of additional experiments.

1. What is the pathogenic relevance of Tat palmitoylation?

- Currently the manuscript provides insights into how palmitoylation regulates Tat-dependent cell biology, but it is less clear under what situations/stages of HIV infection and/or disease pathogenesis palmitoylation is functionally important. The authors suggest that lack of palmitoylation of C31 may explain the reduced neuropathogenicity of a specific HIV-1 clade, Clade C, compared to Clade B. However, the reference cited (Mishra et al., 2008) shows that a C30S mutant (which Fig 1A of the current study shows is normally palmitoylated) can mimic differences between the B and C clades in multiple cell-based assays. Moreover, a second study (Ranga et al., 2004) suggests that C30S and C31S mutants are both similarly defective in a monocyte migration assay, despite the fact that the current study shows that one of these mutants is palmitoylated and the other is not. Based on these findings, the simplest explanation for differences between the B and C clades would appear to be that the CC motif is critical for neuropathogenicity, but that palmitoylation of C31 is not. It is important that the authors more fully discuss the papers referenced above and also perform experiments to more definitively address the functional importance of palmitoylation for viral pathogenicity.

2. What is the mechanism of CypA-dependent palmitoylation of Tat?

- Is this a direct mechanism of CypA on Tat or an indirect mechanism of CypA on DHHC20 or some other protein? More experimental evidence should be provided to address this point. The authors mention that CypA interacts directly with Tat but do not show the data - these data should be presented, in addition to experiments that address how direct such regulation may be (e.g. are specific proline sites in Tat a relevant substrate for CypA?).
- A related question is why pharmacological inhibition or genetic knockdown of either CypA or FKBP12 completely abolishes Tat palmitoylation. Why do these two prolyl isomerases not compensate for each other, at least partially? The authors should provide a more definitive mechanistic explanation (which may again be aided by further experiments) as to what is happening here.

Other points:

1. All western blot images appear highly processed and are not acceptable in their current format. Hopefully the authors still have the raw unprocessed images, which would be more appropriate to use in the figures, or have remaining samples to re-generate images with less processing.
2. In the experiments using Tat-C31O it would be important to know if this recombinant Tat is structurally sound and correctly folded. Can the LTR-driven luciferase assay be performed using exogenously applied Tat-C31O (i.e. the unpalmitoylated form) to address this issue?
3. On line 107 "DDHC-5" should be "DHHC5" and line 143 "plasmalemma" is better written as "plasma membrane".
4. In experiments where there are more than two groups a t-test is not appropriate, instead an ANOVA should be used and in experiments where there are two independent variables (e.g. construct and treatment) a 2-way ANOVA should be used.
5. In figures 3d and 6d the exact number of replicates should be given.
6. There is no mention of replication in figures 1 and 2, These experiments should be replicated at least 3 times.
7. Figure 2c lacks a HAM- control and should be modified to include one.
8. Western blot images should be shown in Figure 3a as it is important to show the protein levels as well as the palmitoylation level for this type of labeling experiment.
9. A number of times the authors state "data not shown". All relevant data should be presented in the main manuscript figures or in supplemental figures.
10. On page 5 of the Results the authors state that virulence of the clade C is essentially the same as other clades, but on page 14 of the Discussion they state that the clade C is of lower virulence. This discrepancy should be addressed.
11. In the Discussion the authors state that palmitoylation might affect progression to latency (page 14, line 321). How might this happen? This does not require more experiments but more discussion than this one sentence would be helpful.

Reviewer #3 (Remarks to the Author):

Nature Communications manuscript #NCOMMS-17-18539, by Chopard et al., presents an excellent analysis of the Tat palmitoylation and its consequences with respect to HIV-1 replication. The authors demonstrate that Tat is palmitoylated on residue C31 by DHHC-20, that palmitoylation requires PI(4,5)P2 binding and the polyisomerases FKBP12 and Cyclophilin A, that Gag expression reduces Tat palmitoylation by virtue of Cyclophilin A depletion, and that palmitoylated Tat perturbs PI(4,5)P2-dependent membrane trafficking processes. These data and the techniques employed are likely to be of interest to a wide audience. I have only a few minor comments:

1. Line 79: It would be nice to see the ITC results in the supplementary data.
2. Line 111: Without some notion of the percentage of cells transfected, and fluorescence microscopy exposure times and gain settings, it is difficult to assess the claim that DHHC proteins were expressed to the same extents in PC12 cells (Supplementary Figure 2).
3. Line 127: Please show some evidence that Tat31O was acylated at 50% efficiency. This is

important because the number was used to substantiate the claim that palmitoylation increases the Tat affinity for PI(4,5)P2 by 5-fold, rather than about 1.5-fold (if acylation were 100%).

4. Figure 2: In Panel b, the cytoplasmic localization of 31S is not so convincing. In Panel c, the preference for palmitoylation by DHHC-20 is not so convincing either. Can better images be provided?

5. The authors argue that palmitoylated Tat perturbs PI(4,5)P2-dependent trafficking processes such as stimulated GH release in PC12 cells, and phagocytosis in MDMs. Have the investigators performed similar experiments with their PH-PLCdelta-EGFP chimeric protein? If so, what were the results?

Rebuttal letter Chopard et al. (NCOMMS-17-18539)

The reviewers' comments appear in *italics*, our responses are in roman.

***Answers to Reviewer #1:**

The authors confirm aspects of their findings in Jurkat, HEK293, and PC12 cell lines, which is dependent on DHHC20, using primary PBMCs and chromaffin cells, but do not examine the role of Tat palmitoylation in primary neurons. PC12 cells can be considered preneurons and poor surrogates of primary neurons. Is Tat Palmitoylated in primary neurons (as shown in Supplementary Table 1) and in sufficient amounts to cause neurotoxicity?

We agree that it was important to validate our findings on primary neurons. It is difficult to obtain sufficient primary neurons to perform biochemistry studies such as immunoprecipitation and MS analysis as for supplementary Table 1. We instead used a cell biology approach to examine whether Tat is palmitoylated in these cells. We observed that Tat accumulates at the plasma membrane of primary neuron cell bodies and somewhat in dendrites / axons. Tat is displaced toward the cytosol by the C31S mutation or upon addition of the palmitoylation inhibitor 2-Bromo palmitate (New supplementary Fig.3). These results were identical to those obtained on PC12 cells (Fig.2b) and indicate that Tat is also palmitoylated on Cys31 in primary neurons and that this modification is required for stable Tat localization at the plasma membrane.

Page 8, lines 140-143; the sentence talking about biphasic membrane localization using Tat/F-actin colocalization is confusing and needs rewriting.

The sentence has been clarified following the reviewer's comment.

Page 9, lines 193-195; the sentence referring to Tat ownership of Pro residues is unclear and needs rewriting.

This point has also been rewritten accordingly.

Although not incorrect, rarely used terms such as in cellulo might be replaced with "in living cells".

We removed the "in cellulo".

***Answers to Reviewer #2**

1. What is the pathogenic relevance of Tat palmitoylation?

• Currently the manuscript provides insights into how palmitoylation regulates Tat-dependent cell biology, but it is less clear under what situations/stages of HIV infection and/or disease pathogenesis palmitoylation is functionally important. The authors suggest that lack of palmitoylation of C31 may explain the reduced neuropathogenicity of a specific HIV-1 clade, Clade C, compared to Clade B. However, the reference cited (Mishra et al., 2008) shows that a

C30S mutant (which Fig 1A of the current study shows is normally palmitoylated) can mimic differences between the B and C clades in multiple cell-based assays. Moreover, a second study (Ranga et al., 2004) suggests that C30S and C31S mutants are both similarly defective in a monocyte migration assay, despite the fact that the current study shows that one of these mutants is palmitoylated and the other is not. Based on these findings, the simplest explanation for differences between the B and C clades would appear to be that the CC motif is critical for neuropathogenicity, but that palmitoylation of C31 is not. It is important that the authors more fully discuss the papers referenced above and also perform experiments to more definitively address the functional importance of palmitoylation for viral pathogenicity.

Mishra et al (2008) showed that addition of Tat-C30S to primary neurons failed to reproduce the numerous effects of exogenous Tat on primary neurons.

An important experimental difference between our original findings and the Mishra et al (2008) paper precludes direct comparison since transfected and exogenous recombinant proteins were used, respectively.

To compare these results it was thus necessary to examine whether recombinant, exogenous Tat-C30S became palmitoylated. We thus prepared recombinant Tat-C30S (for immunofluorescence experiments) and Tat-C30S-His6 (for palmitoylation experiments). We first noticed that this mutant did not strongly bind to the Heparin column (a polyanionic matrix) that we use during recombinant Tat purification. Contrarily to WT Tat, Tat-C30S started to elute at low NaCl concentrations. This is reminiscent of mutants in Tat basic domain and this first observation indicates that Tat basic domain is not properly exposed in Tat-C30S. This phenotype was confirmed when we prebound this mutant to cells: Tat-C30S poorly bound to PC12 cells (only 57% of WT Tat binding). We also observed that Tat-C30S is not transported to the cytosol and therefore not palmitoylated (supplementary Fig.17). Hence, Tat-C30S shows a strong defect in cell binding/ endocytosis and /or translocation to the cytosol. This is why extracellular Tat-C30S failed to become palmitoylated (supplementary Fig.17), while it is palmitoylated following transient transfection (Fig.2a). Our data are consistent with those of Mishra et al because extracellular Tat-C30S was inactive in all the assays they used and actually even less active than Tat-C31S in a few assays.

Hence, recombinant Tat-C30S and Tat-C31S mutants do not enable to examine the respective implication of palmitoylation and chemokine-like effect of Tat in neurotoxicity. It is not possible to conclude at this point whether Tat neuropathogenicity is due to one and / or the other. This is now stated in the Discussion. Further studies using more selective tools are needed to examine this issue.

2. What is the mechanism of CypA-dependent palmitoylation of Tat?

• Is this a direct mechanism of CypA on Tat or an indirect mechanism of CypA on DHHC20 or some other protein? More experimental evidence should be provided to address this point. The authors mention that CypA interacts directly with Tat but do not show the data - these data

should be presented, in addition to experiments that address how direct such regulation may be (e.g. are specific proline sites in Tat a relevant substrate for CypA?).

We agree that the Tat-CypA interaction is an important point. We now present the corresponding data. We found using Tat immunoprecipitation and GST pull downs using GST-CypA that CypA interacts both with Tat and DHHC-20 (New Fig.7). We also observed that Tat specifically interacts with DHHC-20 but not DHHC-5, in agreement with the identification of DHHC-20 as the Tat-palmitoylating enzyme. These two approaches showed that Tat, CypA and DHHC-20 form a complex.

It is however difficult to know if the CypA requirement for the palmitoylation process is due to Tat and/or DHHC20, especially since they both own a potential CypA binding site (see below). CypA probably facilitates Tat palmitoylation via interaction both with DHHC20 and Tat.

Regarding FKBP12, Tat-FKBP12 interaction seems to be weaker since it was only observed using Tat immunoprecipitation (Supplementary Fig.13) and not GST pull-down by GST-FKBP12.

We now discuss how could CypA interact with Tat, *i.e.* which Tat-Prolines could be recognized by CypA (also see the answer to the next point).

[redacted]

• A related question is why pharmacological inhibition or genetic knockdown of either CypA or FKBP12 completely abolishes Tat palmitoylation. Why do these two prolyltransferases not compensate for each other, at least partially? The authors should provide a more definitive mechanistic explanation (which may again be aided by further experiments) as to what is happening here.

The specificity of immunophilins is known to be rather low¹. This is consistent with their role as generalist chaperones. There is nevertheless some substrate specificity that is, at least for CypA and FKBP12, essentially due to the residue before the Pro. Hence, CypA prefers small residues, with the GP motif being the most efficient according to both peptide binding experiments² and structural analysis³. This motif is the one present in HIV-1 capsid protein⁴. This CypA specificity is complementary to that of FKBP12 that prefers large hydrophobic residues such as Leu or Phe before the Pro⁵. Hence there is no sequence specificity for these chaperones but a selection according to the size of the residue before the Pro, and there is minimum overlap between CypA and FKBP12. This point is now discussed in the manuscript.

Regarding Tat, a candidate site for CypA binding is Gly83-Pro84 but it is not very well conserved among viral isolates⁶. DHHC20 also owns a GP motif (residues 253-254). This point is now discussed. For FKBP12 binding to Tat there is no clear candidate site really. A problem with Tat is that most residues before the prolines are charged and this type of peptide has not been tested for CypA or FKBP12 binding.

Other points:

1. All western blot images appear highly processed and are not acceptable in their current format. Hopefully the authors still have the raw unprocessed images, which would be more appropriate to use in the figures, or have remaining samples to re-generate images with less processing.

We have followed conventional recommendations regarding image processing⁷. Luminosity/contrast had to be corrected especially because Biotin blots tend to produce diffusing bands. Following the reviewer's comment we came back to original TIFF files and now provide less contrasted images.

2. In the experiments using Tat-C31O it would be important to know if this recombinant Tat is structurally sound and correctly folded. Can the LTR-driven luciferase assay be performed using exogenously applied Tat-C31O (i.e. the unpalmitoylated form) to address this issue?

Tat-C31O exhibits correct affinity for PIP2, indicating that this mutant maintains some kind of Tat native folding. Nevertheless, each Tat Cys, except C31, is needed for Tat transactivation⁸ and Tat-C31O cannot therefore transactivate.

3. On line 107 "DDHC-5" should be "DHHc5" and line 143 "plasmalemma" is better written as "plasma membrane".

This has been corrected.

4. In experiments where there are more than two groups a t-test is not appropriate, instead an ANOVA should be used and in experiments where there are two independent variables (e.g. construct and treatment) a 2-way ANOVA should be used.

ANOVA is effectively the recommended test when more than two groups are compared. We thus performed these tests when appropriate, using Graphpad Prism, Dunnet post-test for one-way ANOVA and Bonferroni post-test for two-way ANOVA. The results are now included. We obtained similar significance levels compared with the t-tests that we used initially.

5. In figures 3d and 6d the exact number of replicates should be given.

This is now indicated (n=3).

6. There is no mention of replication in figures 1 and 2, These experiments should be replicated at least 3 times.

These experiments were repeated three times and by different experimenters in different laboratories. Moreover, data from PC12 cells presented in Fig.2b were confirmed using primary neurons (Supplementary Fig.3).

7. Figure 2c lacks a HAM- control and should be modified to include one.

This experiment has been repeated and now additionally includes a control for DHHC overexpression obtained using recently available anti-DHHC antibodies.

8. Western blot images should be shown in Figure 3a as it is important to show the protein levels as well as the palmitoylation level for this type of labeling experiment.

These images are now shown as requested.

9. A number of times the authors state "data not shown". All relevant data should be presented in the main manuscript figures or in supplemental figures.

All data previously indicated as not shown are now presented in Fig. 7 or supplemental Figures. Below is the list of these new Figures.

Page (1st version)	Data previously not shown	Figure
5	ITC of Tat-C31S	Supplementary Fig.2
7	Acylation efficiency of Tat-C31O	Supplementary Fig.9
8	Secretion data for chromaffin cells	Supplementary Fig.11
13	Viral production for pNL4.3 Δ Tat + pBi-Tat	Supplementary Fig.15
13	NL4.3-CA-G89V virus encapsidates FKBP12	Supplementary Fig.16
14	Tat-CypA interaction	Fig.7
14	Tat-FKBP12 interaction	Supplementary Fig.13
16	Purity of Tat-C31O	Supplementary Fig.9

10. On page 5 of the Results the authors state that virulence of the clade C is essentially the same as other clades, but on page 14 of the Discussion they state that the clade C is of lower virulence. This discrepancy should be addressed.

We agree with the reviewer that this aspect was effectively not clear. In fact in page 14 we only wanted to mention the lower neurotoxicity of clade C. We clarified this point in the revised paper.

11. In the Discussion the authors state that palmitoylation might affect progression to latency (page 14, line 321). How might this happen? This does not require more experiments but more discussion than this one sentence would be helpful.

Following the reviewer's comment we preferred to remove this speculative comment since there is little experimental evidence to discuss this point.

***Answers to Reviewer #3**

1. Line 79: It would be nice to see the ITC results in the supplementary data.

As also requested by reviewer #2 these results are now shown in supplemental Fig. 2.

2. Line 111: Without some notion of the percentage of cells transfected, and fluorescence microscopy exposure times and gain settings, it is difficult to assess the claim that DHHC proteins were expressed to the same extents in PC12 cells (Supplementary Figure 2).

This comment refers to the new Fig.3a. We agree that it was important to monitor DHHC-EGFP overexpression when we examined their effect on Tat palmitoylation. We used recently available anti-DHHC antibodies and found that, despite the fact that DHH5-EGFP was more efficiently expressed than DHHC20-EGFP, the latter enabled 35-fold more efficient palmitoylation (Fig.3a).

3. Line 127: Please show some evidence that Tat310 was acylated at 50% efficiency. This is important because the number was used to substantiate the claim that palmitoylation increases the Tat affinity for PI(4,5)P₂ by 5-fold, rather than about 1.5-fold (if acylation were 100%).

These MALDI/TOF analyses of Tat-C310 and Tat-C310-Palm are now shown in Supplementary Fig.9.

4. Figure 2: In Panel b, the cytoplasmic localization of 315 is not so convincing.

We replaced the image for better visualization of the cytosolic accumulation of Tat-C315. We also performed line scans that may be even more convincing because they are easier to visualize. Please also note that identical data were obtained on primary neurons (Supplementary Fig.3).

In Panel c, the preference for palmitoylation by DHHC-20 is not so convincing either. Can better images be provided?

We repeated this experiment to obtain clearer results and additionally monitored the expression level of DHHC-EGFP using recently available anti-DHHC antibodies.

5. The authors argue that palmitoylated Tat perturbs PI(4,5)P₂-dependent trafficking processes such as stimulated GH release in PC12 cells, and phagocytosis in MDMs. Have the investigators performed similar experiments with their PH-PLCdelta-EGFP chimeric protein? If so, what were the results?

Different groups studied these effects and found that overexpression of EGFP-PH_{PLCδ}, which results in PI(4,5)P₂ masking, inhibit phagocytosis⁹ and secretion by chromaffin cells¹⁰.

References

1. Galat A. Peptidylprolyl cis/trans isomerases (immunophilins): biological diversity--targets--functions. *Curr Top Med Chem* **3**, 1315-1347 (2003).
2. Piotukh K, Gu W, Kofler M, Labudde D, Helms V, Freund C. Cyclophilin A binds to linear peptide motifs containing a consensus that is present in many human proteins. *The Journal of biological chemistry* **280**, 23668-23674 (2005).
3. Howard BR, Vajdos FF, Li S, Sundquist WI, Hill CP. Structural insights into the catalytic mechanism of cyclophilin A. *Nat Struct Biol* **10**, 475-481 (2003).
4. Gamble TR, *et al.* Crystal structure of human cyclophilin A bound to the amino-terminal domain of HIV-1 capsid. *Cell* **87**, 1285-1294 (1996).
5. Tradler T, Stoller G, Rucknagel KP, Schierhorn A, Rahfeld JU, Fischer G. Comparative mutational analysis of peptidyl prolyl cis/trans isomerases: active sites of Escherichia coli trigger factor and human FKBP12. *FEBS letters* **407**, 184-190 (1997).
6. Debaisieux S, Rayne F, Yezid H, Beaumelle B. The Ins and Outs of HIV-1 Tat. *Traffic* **13**, 355-363 (2012).
7. Rossner M, Yamada KM. What's in a picture? The temptation of image manipulation. *The Journal of cell biology* **166**, 11-15 (2004).
8. Jeang KT, Xiao H, Rich EA. Multifaceted activities of the HIV-1 transactivator of transcription, Tat. *J Biol Chem* **274**, 28837-28840. (1999).
9. Botelho RJ, *et al.* Localized biphasic changes in phosphatidylinositol-4,5-bisphosphate at sites of phagocytosis. *J Cell Biol* **151**, 1353-1368 (2000).
10. Holz RW, *et al.* A pleckstrin homology domain specific for phosphatidylinositol 4, 5-bisphosphate (PtdIns-4,5-P₂) and fused to green fluorescent protein identifies plasma membrane PtdIns-4,5-P₂ as being important in exocytosis. *J Biol Chem* **275**, 17878-17885 (2000).

Reviewers' comments:

Reviewer #1 (Remarks to the Author):

The study by Chopard et al. elucidate a novel, cyclophilin A-dependent mechanism of HIV Tat palmitoylation at Cys31. Palmitoylation greatly increases Tat's affinity for PI(4,5)P₂, causing Tat to bind and accumulate in the cell membrane, which hinders viral budding and exocytosis in infected cells, as well as phagocytosis in macrophages. The authors further demonstrate that this mechanism, problematic for infected cells, is suppressed by HIV Gag, which sequesters peptidyl prolyl isomerase A/cyclophilin A and DHHC-20 in infected cells. The findings provide answers to previously unresolved questions regarding Tat. The data explain why extremely low levels of Tat trigger cytotoxicity in bystander but not infected cells, and additionally clarify Tat's numerous actions independent of its role as a transcription factor—especially regarding its role as a cell penetrating protein. The study also provides insight into why Tat derived from clade C HIV, which lacks Cys31, a critical palmitoylation site for PI(4,5)P₂ binding, is less neurotoxic than Tat derived from other HIV clades in which Cys31 is highly conserved.

In the revised version of the manuscript, the authors have carefully and systematically addressed my prior criticisms.

Reviewer #2 (Remarks to the Author):

The revised manuscript by Chopard et al. "Cyclophilin A enables specific HIV-1 Tat palmitoylation and accumulation in uninfected cells" addresses most of the initial concerns of this reviewer. The authors made considerable effort to add new experiments, particularly regarding major concern #1: "What is the pathogenic relevance of Tat palmitoylation?" These new experiments do not fully address the point raised but do clarify the discrepancy between the data presented in this manuscript and those in Mishra et al 2008. The additional discussion, along with the new data, sufficiently addresses this concern. However, there are still minor concerns regarding the revised manuscript as follows:

1. Concern #2 was not fully experimentally addressed. The new interaction data are helpful but do not fully elucidate the underlying mechanism. It would be sufficient to more explicitly state that defining the target(s) of CypA (e.g. prolyl isomerization of a Tat proline, a DHHC20 proline, or a proline of some other protein) will require further investigation.
2. The related point, regarding why CypA and Fkbp12 do not compensate for one another, was not fully addressed. The authors state that the two prolyl isomerases prefer different residues N-terminal to the substrate proline residue. This suggests that CypA and Fkbp12 likely have different targets (i.e. they either act on different residues of the same protein, or on different proteins). The updated discussion on p16 does not add much mechanistic insight regarding this point (e.g. the statement "Tat possesses several Pro residues" would be true for almost all proteins) and should be more focused or removed. As with point #1 above, the authors should more explicitly state the lack of a known mechanism for this

observation.

3. The images in figure 1c appear to have been stretched and cropped compared to those in the original submission such that the "HA-" and "HA+" indicators no longer mark the appropriate lanes. This should be corrected.

4. In figure 2a the authors state that multiple films were assembled to generate the full panel. Break points between the films should be indicated.

5. The new Fig 3b is interesting but raises technical concerns because the biotin (i.e. palmitoylation-dependent) signal is much higher in the negative control ('-HA') lanes than in other experiments e.g. panel 'c' of the same Figure. The biotin signal thus appears non-specific. This experiment should be repeated. A 'no DHHC' control should also be included and the relative effect of each PAT compared to 'no DHHC' should be quantified (at present it is unclear what the quantitation in the adjacent histogram is plotted relative to). The experiment should be performed at least 3 times and error bars should be indicated on the resulting graph.

6. Similarly, on Figure 7b, error bars should be indicated on the graph.

7. In supplemental figure 13, a panel showing the amount of Tat-FLAG immunoprecipitated and the input levels of all proteins should be shown.

8. A better quality image should be provided for supplemental figure 15.

Reviewer #3 (Remarks to the Author):

The revised version of the Chopard et al. manuscript, "Cyclophilin A enables specific HIV-1 Tat palmitoylation and accumulation in infected cells," represents a modest improvement on the original. The observations are still interesting, and likely to attract a general audience; and the results are generally convincing. I was somewhat disappointed that certain concerns were not addressed more thoroughly, and that some claims were made without pertinent caveats. Specific notes are as follows:

1. Figures 2b and Supplementary Figure 3: These figures were presented to show that the Tat C31S mutation causes the protein to localize cytoplasmically, and that Tat palmitoylation is required for membrane association in neurons. I believe these are valid conclusions. Unfortunately, the authors have not made sufficient efforts to quantify their data. For at least a dozen cells of each condition, it would be good to see quantitative colocalization analyses with cytoplasmic and/or plasma membrane markers. Alternatively, the authors might perform line plot analyses as in Figure 2b, but normalizing for cell width and pixel brightness, and summing at least a dozen plots of each cell type. As it is, the line plot choice for the one 31S cell analyzed in Figure 2b (bottom right) does not seem chosen objectively to cross the full width of the cell, but to support a point.

2. Figure 3a: This new figure is a concern. At face value, it suggests that endogenous PC12 levels of DHHC20 are very high relative to the levels of DHH5. Given the marginal increase in total DHHC20 levels (DHHC20 plus DHHC20-GFP) in the DHHC20-GFP transfections, why do we see any increase in transfected Tat palmitoylation relative to untransfected cells or to cells transfected with DHHC5-GFP (which still are expected to have high levels of endogenous DHHC20)?

3. Supplementary Figure 9: I don't understand why the m/z difference for palmitoylated Tat

244 Daltons, rather than the predicted mass increase of 238 Daltons. Please explain. Additionally, MALDI-TOF is often thought to work poorly for quantitation, especially in the absence of internal standards. A statement as to the degree of certainty or uncertainty of the percent palmitoylated Tat should be included.

Rebuttal letter Chopard *et al.* (NCOMMS-17-18539A)

The reviewers' comments appear in *italics*, our responses are in roman.

*Answers to Reviewer #1

There were no further comments.

*Answers to Reviewer #2

1. Concern #2 was not fully experimentally addressed. The new interaction data are helpful but do not fully elucidate the underlying mechanism. It would be sufficient to more explicitly state that defining the target(s) of CypA (e.g. prolyl isomerization of a Tat proline, a DHHC20 proline, or a proline of some other protein) will require further investigation.

The discussion has been modified as suggested by the Reviewer (lines 398-401).

2. The related point, regarding why CypA and Fkbp12 do not compensate for one another, was not fully addressed. The authors state that the two prolyl isomerases prefer different residues N-terminal to the substrate proline residue. This suggests that CypA and Fkbp12 likely have different targets (i.e. they either act on different residues of the same protein, or on different proteins). The updated discussion on p16 does not add much mechanistic insight regarding this point (e.g. the statement "Tat possesses several Pro residues" would be true for almost all proteins) and should be more focused or removed. As with point #1 above, the authors should more explicitly state the lack of a known mechanism for this observation.

The statement "Tat possesses several Pro residues" was removed from the Discussion and we now indicate that the identity of CypA and FKBP12 targets in the palmitoylation complex remain to be identified (lines 398-401).

3. The images in figure 1c appear to have been stretched and cropped compared to those in the original submission such that the "HA-" and "HA+" indicators no longer mark the appropriate lanes. This should be corrected.

We apologize for this mistake which has been corrected.

4. In figure 2a the authors state that multiple films were assembled to generate the full panel. Break points between the films should be indicated.

The break point between the two blots is now indicated by a dashed line in Fig.2A.

5. The new Fig 3b is interesting but raises technical concerns because the biotin (i.e. palmitoylation-dependent) signal is much higher in the negative control ('-HA') lanes than in other experiments e.g. panel 'c' of the same Figure. The biotin signal thus appears non-specific. This

experiment should be repeated. A 'no DHHC' control should also be included and the relative effect of each PAT compared to 'no DHHC' should be quantified (at present it is unclear what the quantitation in the adjacent histogram is plotted relative to). The experiment should be performed at least 3 times and error bars should be indicated on the resulting graph.

This experiment with PC12 rat-cells was problematic. In fact, the main difficulty we faced is that, for some reason, transfection with mouse DHHC20-GFP inhibited endogenous DHHC20 expression (previous Fig3a). This was not the case for DHHC5, and this effect complicated the interpretation of this experiment. We thus repeated this experiment using HEK human cells transfected with human DHHC enzymes (myc tagged), and to detect palmitoylation we used the Acyl-resin assisted capture (Acyl-RAC) technique that is simpler and more sensitive than ABE¹. It is also more easily quantitative². We obtained data (triplicate blots + quantification as requested, new Fig3.a-b) that identified DHHC20 as the Tat palmitoylating enzyme.

6. Similarly, on Figure 7b, error bars should be indicated on the graph.

Error bars are now presented as requested (n= 3 experiments).

7. In supplemental figure 13, a panel showing the amount of Tat-FLAG immunoprecipitated and the input levels of all proteins should be shown.

We performed a new experiment that included these important controls and the requested panels are now shown.

8. A better quality image should be provided for supplemental figure 15.

For this experiment, whose aim was to verify that cotransfection with Tat plasmids and Δ Tat viruses enabled viral production, it was important to obtain quantitative data. When we repeated the experiment, we therefore used a p24 ELISA instead of Western blot to monitor viral production. We now show both ELISA and Western blot data.

***Answers to Reviewer #3**

1. Figures 2b and Supplementary Figure 3: These figures were presented to show that the Tat C31S mutation causes the protein to localize cytoplasmically, and that Tat palmitoylation is required for membrane association in neurons. I believe these are valid conclusions. Unfortunately, the authors have not made sufficient efforts quantify their data. For at least a dozen cells of each condition, it would be good to see quantitative colocalization analyses with cytoplasmic and/or plasma membrane markers. Alternatively, the authors might perform line plot analyses as in Figure 2b, but normalizing for cell width and pixel brightness, and summing at least a dozen plots of each cell type. As it is, the line plot choice for the one 31S cell analyzed in Figure 2b (bottom right) does not seem chosen objectively to cross the full width of the cell, but to support a point.

We agree with the reviewer that it is more robust to follow plasma membrane localization using colocalization with a plasma membrane marker instead of line plots as we did initially.

We thus followed the localization of Tat-mutants to the plasma membrane by quantifying Tat / F-actin colocalization. This was done for all Tat mutants and using $n > 10$ cells as suggested. The corresponding graph is now shown in Fig. 2C.

Please note that Tat displacement to the cytosol caused by the C31S mutation was obtained both on PC12 and primary neurons by two different experimenters in different laboratories (one in Montpellier, the other in Strasbourg) and using different neuron preparations.

2. Figure 3a: This new figure is a concern. At face value, it suggests that endogenous PC12 levels of DHHC20 are very high relative to the levels of DHH5. Given the marginal increase in total DHHC20 levels (DHHC20 plus DHHC20-GFP) in the DHHC20-GFP transfections, why do we see any increase in transfected Tat palmitoylation relative to untransfected cells or to cells transfected with DHHC5-GFP (which still are expected to have high levels of endogenous DHHC20)?

We agree, this figure has been replaced (new Fig.3a and b). Please see our detailed answer to Reviewer #2, point 5.

3. Supplementary Figure 9: I don't understand why the m/z difference for palmitoylated Tat 244 Daltons, rather than the predicted mass increase of 238 Daltons. Please explain.

This difference indeed looked puzzling. We contacted the persons who performed this Mass Spectroscopy (MS) analysis, and they answered that they apologized for not pointing out that, with this MS instrument, masses are provided ± 3 Da. Hence $244 \pm 3 \sim 238 \pm 3$ Da. The figure was corrected accordingly.

Additionally, MALDI-TOF is often thought to work poorly for quantitation, especially in the absence of internal standards. A statement as to the degree of certainty or uncertainty of the percent palmitoylated Tat should be included.

There are two types of MS-based proteomics. The conventional "bottom-up" approach involves protein digestion into short peptides before analysis by MS. This approach is not suitable for the study of post-translational modifications (PTMs) because only a small and variable fraction of peptides is recovered^{3,4}. Here we used a "Top-down" approach involving direct analysis of the entire protein. This top-down MS approach is well suited for PTM study because PTM effect on the physico-chemical properties of the intact protein is usually negligible compared with that of the peptides and, as far as the relative abundance of protein species is studied, this method is thought to have a precision of $< 5\%$ ^{4,5}. Precision is nevertheless depending on the quality of the MS spectra. Here unmodified Tat-C31O is the internal standard and we calculated that Tat-C31O palmitoylation took place with an efficiency of $47 \pm 8 \%$. The figure has been modified accordingly.

References

1. Edmonds MJ, Geary B, Doherty MK, Morgan A. Analysis of the brain palmitoyl-proteome using both acyl-biotin exchange and acyl-resin-assisted capture methods. *Sci Rep* **7**, 3299 (2017).
2. Werno MW, Chamberlain LH. S-acylation of the Insulin-Responsive Aminopeptidase (IRAP): Quantitative analysis and Identification of Modified Cysteines. *Sci Rep* **5**, 12413 (2015).
3. Siuti N, Kelleher NL. Decoding protein modifications using top-down mass spectrometry. *Nature methods* **4**, 817-821 (2007).
4. Zhang H, Ge Y. Comprehensive analysis of protein modifications by top-down mass spectrometry. *Circ Cardiovasc Genet* **4**, 711 (2011).
5. Pesavento JJ, Mizzen CA, Kelleher NL. Quantitative analysis of modified proteins and their positional isomers by tandem mass spectrometry: human histone H4. *Anal Chem* **78**, 4271-4280 (2006).

REVIEWERS' COMMENTS:

Reviewer #2 (Remarks to the Author):

All concerns of this reviewer are now satisfactorily addressed.

Reviewer #3 (Remarks to the Author):

The newly modified version of the manuscript NCOMMS-17-18539B, "Cyclophilin A enables specific HIV-1 Tat palmitoylation and accumulation in uninfected cells," has addressed my previous concerns. At this point it would seem acceptable for publication.